# Vintix: Action Model via In-Context Reinforcement Learning

**Andrei Polubarov** [* 1 2 3] **Nikita Lyubaykin** [* 1 4] **Alexander Derevyagin** [* 1 5] **Ilya Zisman** [1 2 3] **Denis Tarasov** [1]
**Alexander Nikulin** [1 6] **Vladislav Kurenkov** [1 4]

## Abstract

In-Context Reinforcement Learning (ICRL) represents a promising paradigm for developing generalist agents that learn at inference time through trial-and-error interactions, analogous to how large language models adapt contextually, but with a focus on reward maximization. However, the scalability of ICRL beyond toy tasks and single-domain settings remains an open challenge. In this work, we present the first steps toward scaling ICRL by introducing a fixed, cross-domain model capable of learning behaviors through in-context reinforcement learning. Our results demonstrate that Algorithm Distillation, a framework designed to facilitate ICRL, offers a compelling and competitive alternative to expert distillation to construct versatile action models. These findings highlight the potential of ICRL as a scalable approach for generalist decision-making systems. Code to be released at dunnolab/vintix.

## 1. Introduction

The pursuit of generalist control and decision-making agents has long been a major target of the reinforcement learning (RL) research community (Sutton & Barto, 1998). These agents are envisioned to handle a diverse set of tasks while exhibiting adaptation properties such as self-correction and self-improvement based on reward functions. Notably, traditional online RL algorithms demonstrate these properties in narrow domains and can achieve exceptional performance (Berner et al., 2019; Badia et al., 2020; Schrittwieser et al., 2020; Baker et al., 2022; Team et al., 2023). However, their online nature of learning and frequent reliance on environment-specific training is still considered a challenge to overcome limiting the scalability (Dulac-Arnold et al.,

*Equal contribution [1]AIRI [2]Skoltech [3]Research Center for Trusted Artificial Intelligence, ISP RAS [4]Innopolis University [5]HSE [6]MIPT. Correspondence to: Vladislav Kurenkov <kurenkov@airi.net>. Work done by dunnolab.ai.

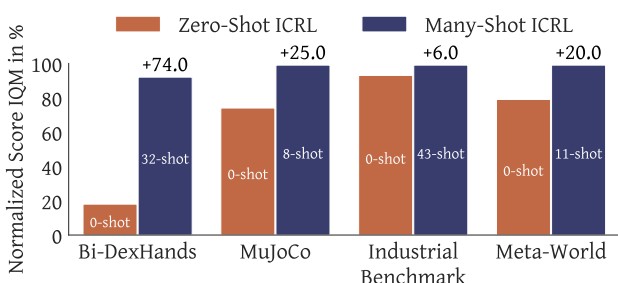

*Figure 1.* **Vintix: Self-Correction with Many-Shot ICRL.** Many-shot ICRL consistently self-corrects online on the training tasks (87) nearing the demonstrators' performance across four domains. Optimal number of shots for many-shot ICRL are shown inside the bar for each task. One shot corresponds to one online episode. See Section 3 for more details and comparisons to action models based on expert distillation.

2019; Levine et al., 2020).

In contrast to online reinforcement learning, a recent boom of generative models, revitalized by Large Language Models and their emergent properties (Brown et al., 2020), sparked a high amount of interest in leveraging offline data for RL training: from specialized offline RL algorithms (Levine et al., 2020; Tarasov et al., 2024) to *Action Models* capable of handling many cross-domain tasks at the same time (Reed et al., 2022; Gallouédec et al., 2024; Haldar et al., 2024; Collaboration et al., 2024; Schmied et al., 2024; Sridhar et al., 2024).

Current approaches to pre-training cross-domain action models broadly fall into two categories. The first utilizes all available data, conditioning policies on return-to-go targets in a framework inspired by upside-down reinforcement learning principles (Schmidhuber, 2019; Lee et al., 2022; Schmied et al., 2024). The second, and currently dominant, approach prioritizes expert demonstrations, often disregarding reward signals entirely, as exemplified by Reed et al. (2022) and Haldar et al. (2024).

While the field increasingly steers toward language and demonstration-guided paradigms over reward-centric approaches, the "Reward is enough" principle (Silver et al., 2021) continues to offer a foundational framework for

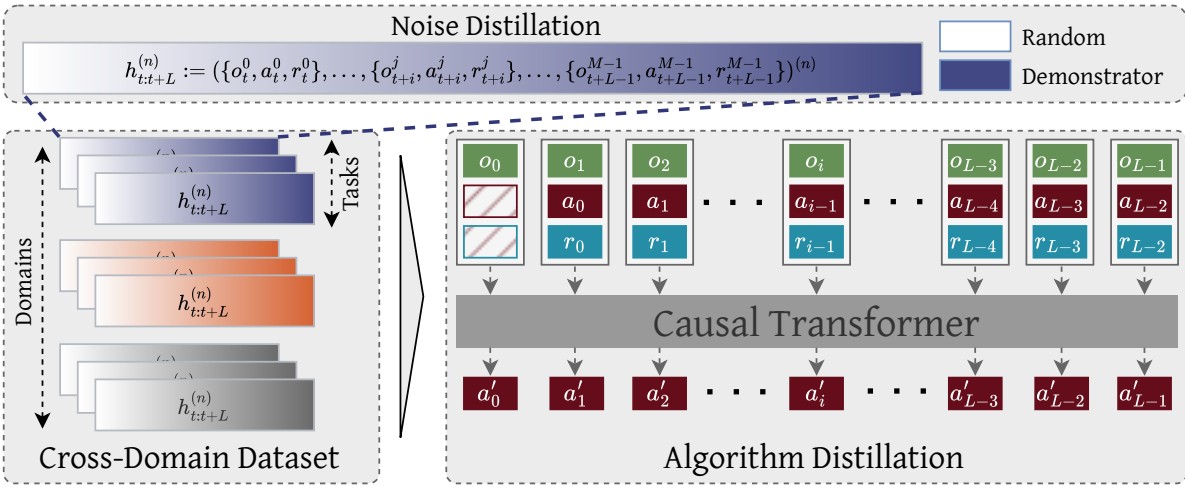

*Figure 2.* **Vintix: Approach Overview.** Stage 1 (Noise Distillation) - approximating policy improvement trajectory by injecting gradually annealed uniform noise (see Algorithm 1). Stage 2 (Cross-Domain Dataset) - combining collected multi-episodic sequences into shared cross-domain dataset for subsequent model training. Stage 3 (Algorithm Distillation) - running AD (Laskin et al., 2022) with collated $\{s, a, r\}$ triplets on collected dataset.

agents that learn to maximize arbitrary rewards through trial and error. This perspective has inspired methods like Algorithm Distillation (Laskin et al., 2022), which operationalizes reward-driven learning by training agents via next-token prediction over historical reinforcement learning (RL) trajectories — a mechanism that directly enables In-Context Reinforcement Learning (ICRL). Though initial implementations of this framework, alongside extensions such as by Zisman et al. (2024a;b); Sinii et al. (2024); Nikulin et al. (2024), have been demonstrated only on toy and grid-based tasks within single domains, their data-driven origin positions them as a promising avenue for scaling reward-centric agents to broader, more complex environments.

In this work, we start an investigation into cross-domain action models through In-Context Reinforcement Learning (ICRL), grounded in the Algorithm Distillation framework (Laskin et al., 2022). By focusing on a data-centric approach to ICRL (summarized in Figure 2), we present Vintix, a multi-task action model that exhibits initial signs of inference-time self-correction on training tasks (Figure 1) and preliminary evidence of adaptation to parametric task variations (Section 3.3).

Our key contributions are as follows:

- **Continuous Noise Distillation** (Section 2.1): We propose a data collection strategy extending the work of Zisman et al. (2024a) to continuous action spaces to ease the collection of training data.

- **Open Tools and Datasets for ICRL** (Section 2.2): We publicly release datasets for 87 tasks across

four domains (Meta-World, MuJoCo, Bi-DexHands, Industrial-Benchmark), along with data collection tools and instrumentation to support the development of action models eliciting ICRL behavior.

- **Cross-Domain Scaling of ICRL** (Section 3): We empirically demonstrate that the proposed model, Vintix, can self-correct to attain demonstrator-level performance on training tasks (Figure 1) and adapt to controlled parametric task variations at inference-time.

## 2. Approach

At the core of our approach (Figure 2) is Algorithm Distillation (Laskin et al., 2022), a two-step Offline Meta-RL algorithm. The first step involves collecting ordered training histories from base reinforcement learning (RL) algorithms, while the second step involves a decoder-only transformer trained solely for the next-action prediction. This approach facilitates in-context learning by effectively distilling the policy improvement operator into a causal sequence model. We further propose two augmentations to this technique: (1) democratizing the data collection process by introducing a continuous extension of the noise-distillation procedure by Zisman et al. (2024a); (2) conducting generalist agent-style cross-domain training on the acquired dataset.

### 2.1. Continuous Noise Distillation

Collecting learning histories can be time-consuming and computationally expensive, as it requires curated learning histories of RL algorithms for each task individually. The

resulting trajectories may be excessively long due to poor sample efficiency and may exhibit noise due to training instabilities. Recently, Zisman et al. (2024a) demonstrated that learning trajectories approximated through noise distillation can facilitate the emergence of in-context learning capabilities via next-action prediction. In this framework, called $AD^\epsilon$, policy improvement is approximated by gradually reducing the proportion of random noise injected into the demonstrator policy during its execution in the environment. More specifically, at each time step, a random action is selected with probability $\epsilon$, while a demonstrator action is chosen with probability $1 - \epsilon$. $\epsilon$ is annealed throughout the trajectory, starting at $\epsilon = 1$ (random policy) at the beginning and gradually decreasing to $\epsilon = 0$ (demonstrator policy) by the end.

The $AD^\epsilon$ data collection strategy was originally designed for discrete action spaces. In our work, we propose its extension to continuous action spaces, where the resulting action is defined as a linear mixture of uniform random noise and demonstrator actions. This contrasts with Brown et al. (2019), who employed an epsilon-greedy approach that alternates between fully random and fully expert actions with probability $\epsilon$. The pseudo-code for our noise-distillation procedure is formalized in Algorithm 1.

---

**Algorithm 1** Noise distillation for continuous action spaces

---

**Require:** Demonstrator policy $\pi_{\mathrm{D}}$, task environment, noise schedule $\mathcal{E}$, number of time steps in the trajectory $T$, trajectory buffer $\mathcal{D}$, action space lower and upper bounds $a_{min}, a_{max}$

1: Sample $s_0$ from task environment
2: **for** $t \in T$ **do**
3:     Noise magnitude: $\epsilon_i = \mathcal{E}(t)$
4:     Noise: $u \sim Uniform(a_{min}, a_{max})$
5:     Current action: $a_i = (1 - \epsilon_i) * \pi_{\mathrm{D}}(s_i) + \epsilon_i * u$
6:     Obtain $\{s_{i+1}, r_i, t_i\}$ by executing $a_i$ in task environment
7:     Append $\{s_i, a_i, s_{i+1}, r_i, t_i\}$ to $\mathcal{D}$
8: **end for**

---

The noise schedule $\mathcal{E}$ plays a crucial role in the generation of training trajectories. We observed that linear annealing of $\epsilon$ often results in non-smooth trajectories with abrupt transitions from random to demonstrator performance. This effect can negatively impact the model's convergence; therefore, careful tuning of the noise schedule was necessary for certain tasks. A more detailed description of the epsilon decay functions is provided in the Appendix A.3. Figure 3 illustrates that Algorithm 1 generates trajectories with smooth reward curves, closely resembling the learning histories observed in Zisman et al. (2024a). For Task-Level dataset visualization please refer to Appendix E.

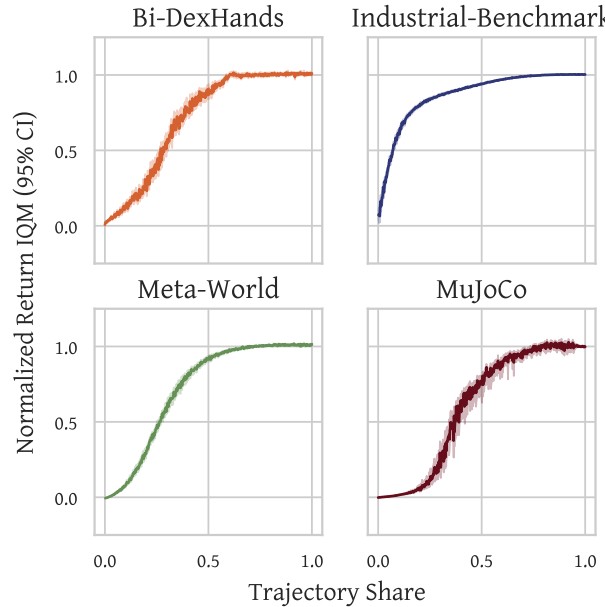

*Figure 3.* **Continuous Noise Distillation trajectories.** Aggregated normalized returns for the collected cross-domain dataset. Returns are normalized with respect to random and demonstrator scores, while trajectory lengths are reported as a fraction of their maximum values.

## 2.2. Cross-Domain Dataset

Building upon Continuous Noise Distillation method, we then collect a cross-domain dataset. Our dataset consists of environments with continuous N-dimensional vector observations and continuous multi-dimensional actions. This focus was chosen to isolate the challenge of processing multi-modal inputs from the challenge of inference-time adaptation across different tasks and domains, allowing us to concentrate on the latter. The collected cross-domain dataset consists of 87 distinct tasks spanning across four domains:

1. **MuJoCo** (Todorov et al., 2012) - classical multi-joint dynamics control suite containing 11 various tasks with different state-action spaces and reward funcions.

2. **Meta-World** (Yu et al., 2021) - benchmark for Multi-Task and Meta RL containing 45 manipulation tasks with shared state and action structure.

3. **Bi-DexHands** (Chen et al., 2022) - benchmark simulator that includes 15 diverse tasks focused on dexterous manipulation.

4. **Industrial-Benchmark** (Hein et al., 2017) - benchmark featuring synthetic continuous control tasks that simulate the properties of real industrial problems.

| Train Dataset | | | | |
|---|---|---|---|---|
| **Domain** | **Tasks** | **Episodes** | **Timesteps** | **Sample Weight** |
| Bi-DexHands | 15 | 216k | 31,7M | 14,2% |
| Industrial-Benchmark | 16 | 96k | 24M | 10,8% |
| Meta-World | 45 | 670k | 67M | 30,1% |
| MuJoCo | 11 | 665k | 100M | 44,9% |
| **Overall** | **87** | **1,6M** | **222,7M** | **100%** |

*Table 1.* **Cross-Domain Dataset summary.** Aggregation is performed by summing all transitions (in the form of $\{s_i, a_i, s_{i+1}, r_i, t_i\}$) across all trajectories collected for all tasks. Detailed dataset statistics can be found ad Appendix G

Table 1 contains detailed statistics of the training dataset. More detailed information on datasets can be found in the Appendix A.

**Training Demonstrators** For MuJoCo and Meta-World we utilized demonstrator policies provided in JAT (Gallouédec et al., 2024). However, for several tasks within Meta-World, we trained our own demonstrators due to the unsatisfactory performance of the provided experts. Bi-DexHands demonstrators were trained using official PPO (Schulman et al., 2017) implementation with an increased number of parallel environments. The demonstrator policies for the Industrial Benchmark were trained using the provided scripts, which were built on top of the Stable-Baselines 3 library (Raffin et al., 2021). In-depth view of demonstrators performance can be found in the Appendix H.

## 2.3. Training and Inference Pipeline

The input data consists of a set of multi-episodic sub-sequences sampled from the original noise-distilled trajectories, formalized as follows:
$h^{(n)}_{t:t+L} := (\{o^0_t, a^0_t, r^0_t\}, ..., \{o^j_{t+i}, a^j_{t+i}, r^j_{t+i}\}, ...$
$, \{o^{M-1}_{t+L-1}, a^{M-1}_{t+L-1}, r^{M-1}_{t+L-1}\})^{(n)}$. Where $t$ is the index of the sub-sequence's starting point within the full noise-distilled trajectory, $L$ is the length of the sub-sequence, $M$ denotes the total number of episodes within the sub-sequence, which varies due to differing episode lengths across tasks, $i \in [0, L-1]$ is a lower subscript timestamp index within the sub-sequence, $j \in [0, M-1]$ is an upper subscript episode index within the sub-sequence. The global subscript $n$ identifies a unique task, which is uniformly sampled from a multi-domain dataset, defined as: $\mathcal{M}_n = \bigcup_{d=1}^{D} \mathcal{M}^d_{n_d}$, where $d \in [1, D]$ represents the domain identifier, and $n_d$ denotes the task belonging to the respective domain. The overall data pipeline closely resembles that of Laskin et al. (2022), with the only difference being its cross-domain coverage.

### 2.3.1. MODEL ARCHITECTURE

At a high level, Vintix consists of three main components: an encoder, which maps raw input sequences $h^{(n)}_{t:t+L}$ into a fixed-size embedding space; a transformer backbone, which processes the encoded inputs; and a decoder, which maps the hidden states produced by the transformer back to the original action space.

All input sequences $h^{(n)}_{t:t+L}$ are split into groups based on observation and action space dimensionalities. For each group, a separate encoder and decoder MLP head are created, enabling the model to map variable observation and action spaces into a shared embedding space. It is important to note that the model is task-agnostic in the sense that it has access only to the dimensionality-based group identifier, but not to an individual task identifier. While the task identifier is a unique ID assigned to each task in the dataset, the group identifier simply indicates whether a set of tasks shares the same observation and action space dimensions, as well as the semantic meaning of each channel.

In contrast to Laskin et al. (2022), which processes each entity in the sequence $h^{(n)}_{t:t+L}$ as a separate token, we stack the representations of the previous action, previous reward, and current observation into a single sequence token. This approach is consistent with token alignment methods proposed by Duan et al. (2016); Grigsby et al. (2024a;b). Such design choice allowed us to significantly expand the context window size by compressing its representation by a factor of three.

In summary, Vintix is a 300M-parameter next-action prediction model with 24 layers, 16 heads, an embedding size of 1024, and a post-attention feed-forward hidden size of 4096. TinyLLama (Zhang et al., 2024) was chosen as the transformer backbone for the Vintix model.

### 2.3.2. TRAINING

The input batches are created by collating together multiple input sequences $h^{(n)}_{t:t+L}$. To standardize the data across tasks, rewards are scaled by task-specific factors (see Appendix G), and observations are normalized. This procedure significantly enhances model performance by reducing task distinguishability based on raw input values, thereby encouraging the model to rely on contextual information to infer the task.

Afterward, sequences are processed by the corresponding encoders to obtain fixed-size representations. The sequence of representations is then passed into a transformer, whose outputs are subsequently decoded into predicted actions $a_{\text{pred}}$. The training objective is to minimize the Mean Squared Error loss between the predicted actions $a_{\text{pred}}$ and the ground-truth trajectory actions $a_{\text{true}}$. Training is conducted on 8

H100 GPUs with a batch size of 64 and 2 gradient accumulation steps. The input sequence length $L$ is set to 8192. For more detailed hyperparameter information, refer to Appendix D.

### 2.3.3. INFERENCE

Model inference is performed iteratively, starting with an empty context. The model generates the first action based on the initial observation and subsequently receives the next observation and reward. Each new transition tuple is appended to the context. When the context exceeds its maximum length $L$, the oldest token is removed, effectively implementing a sliding attention window (Beltagy et al., 2020). To accelerate inference, we utilized the KV-cache implementation from the FlashAttention library (Dao et al., 2022). Since absolute positional encodings are incompatible with sliding attention window with KV-caching due to positional drift, we employed ALiBi encodings (Press et al., 2022), which are inherently relative in nature.

## 3. Results

### 3.1. Inference-Time Self-Correction on Training Tasks

First, we aim to verify whether the Vintix model has the capability for context-based inference-time adaptation through self-correction. To achieve this, we deploy the model on training tasks (ML45 split for Meta-World, ML20 split for Bi-DexHands and setpoints $p \in [0, 75]$ for Industrial-Benchmark) by iteratively unrolling its actions in a cold-start manner, beginning with an empty initial context. Figure 4 illustrates that the model progressively improves its policy in each domain as the number of shots (episodes played) increases. The agent starts with a suboptimal performance and gradually self-corrects by inferring task-related information from the accumulated context, ultimately reaching near-demonstrator-level performance.

Notably, this improvement is both progressive and consistent along the shots-made axis and is observed across multiple domains with varying dynamics and morphology. For task-level graphs depicting inference-time performance with an empty context, refer to Appendix F.

This behavior suggests that, despite being task-agnostic, the model can infer environment's implicit structure and utilize it to enable self-corrective adaptation across training tasks.

### 3.2. Comparison to Related Action Models

Next, we aim to assess the performance of Vintix compared to other generalist agents trained across multiple domains and determine whether its self-corrective inference provides an advantage in matching demonstrator performance levels.

To verify this, we compare the average demonstrator-normalized performance of Vintix with that of Gallouédec et al. (2024) and Sridhar et al. (2024) across training tasks in overlapping domains (MuJoCo and Meta-World). JAT and REGENT scores are taken directly from the respective papers. For Vintix, we extract the average performance over 100 episodes after reaching inference-time convergence (i.e., achieving the best $k$-shot performance per task).

Since most of our demonstrators were sourced from JAT - and REGENT also adopted random and expert scores from JAT - the comparison of normalized scores is valid. However, we improved expert performance for several tasks by fine-tuning hyperparameters and normalized our raw returns with respect to these newly enhanced scores. This adjustment lowers our normalized performance relative to JAT and REGENT, but we adopted this comparison approach as our initial focus is on evaluating the agent's ability to match demonstrator performance rather than comparing absolute scores.

Figure 5 illustrates that inference-time self-correction enables Vintix to outperform JAT and REGENT on Meta-World and MuJoCo by significant margins. More detailed comparison can be found in Appendix I. These results may further highlight a fundamental advantage of Algorithm Distillation over Expert Distillation, namely its adaptability, as initially reported in Laskin et al. (2022).

### 3.3. Generalization Analysis

In this subsection, we take a closer look at the models performance in regards to both unseen parametric variations and tasks.

**Generalization to Parametric Variations**

Further experiments aim to assess whether the Vintix model is capable for context-based inference-time adaptation to task variations that were not encountered during training.

To evaluate this, we performed the cold-start inference procedure described in previous sections on a set of MuJoCo environments with unobservable variations in viscosity and gravity, which were also unseen during training (for more details on MuJoCo parameter variations, refer to Appendix A.2.1). Additionally, we applied the same procedure to unseen environments in the Industrial-Benchmark domain (setpoint $p \in [80, 100]$, see Appendix A.2.4).

Figure 6 presents the experimental results. As shown, Vintix is still able to achieve near-demonstrator performance in modified environments across both domains. However, a slight decline in convergence quality is observed—slower convergence speed in MuJoCo and subtly diminished asymptotic performance in Industrial-Benchmark. The slower convergence rate suggests that the model requires more iterations of self-correction to reach demonstrator-level policy

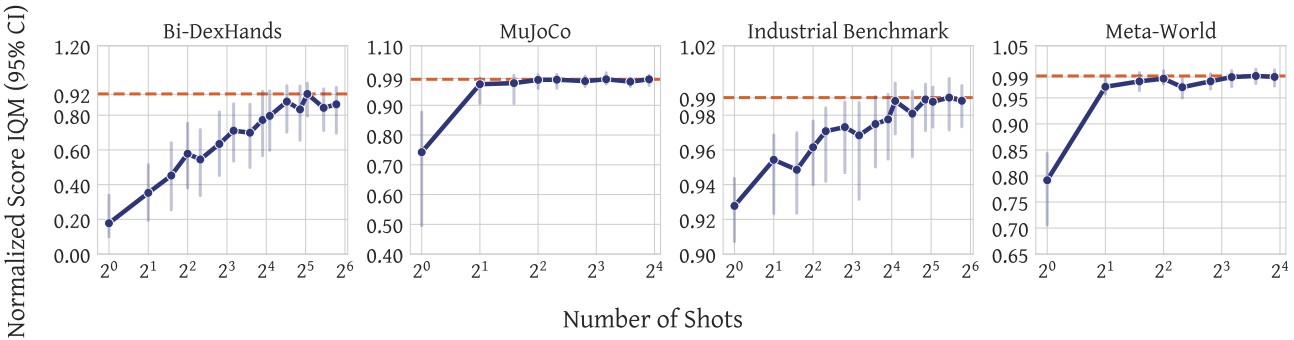

*Figure 4.* **Dynamic self-correction with Many-Shot ICRL.** The model's actions are executed iteratively in the environment without any initial context or task identifier provided to the agent (i.e., cold start). Although it starts with a suboptimal policy, the model gradually improves through context-based self-correction. Results are aggregated over training tasks (ML45 split for Meta-World, ML20 split for Bi-DexHands and setpoints $p \in [0, 75]$ for Industrial-Benchmark) within each domain.

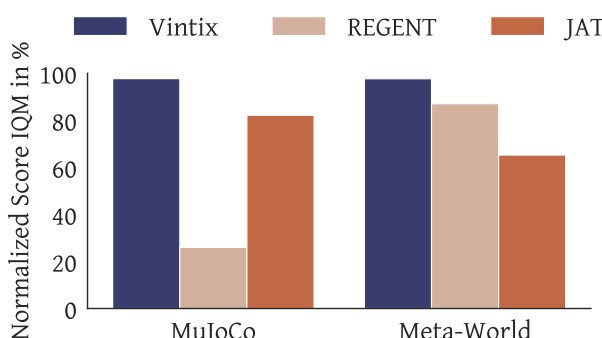

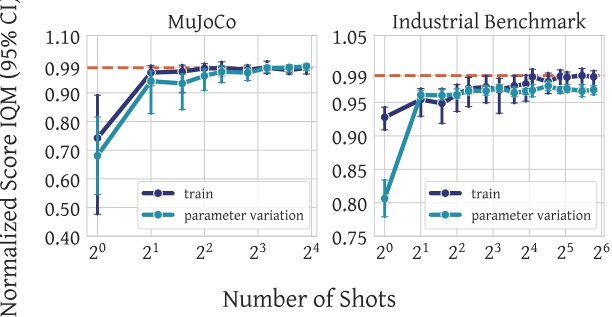

*Figure 5.* Average domain-level demonstrator-normalized returns on training tasks: Vintix vs. REGENT vs. JAT. JAT and REGENT scores are taken directly from the respective papers, while for Vintix, we analyze 100 episodes after reaching inference-time convergence.

*Figure 6.* Cold-start many-shot inference procedure on tasks with parameter variations compared to training tasks. In MuJoCo, variations include changes in viscosity (0.05 and 0.1 vs. the original 0) and gravity ($\pm$ 10%). For the Industrial-Benchmark, we evaluate the model on previously unseen setpoint values, $p \in [80, 100]$.

in environments with moderate parametric variations.

**Generalization to New Tasks**

Finally, we evaluate Vintix's performance on entirely new tasks that were not seen during training to determine whether it can perform in-context reinforcement learning by inferring task structure in this challenging setting.

As in the previous experiments, we unroll actions from Vintix with no initial context on test tasks from the Meta-World ML45 split and the Bi-DexHands ML20 split (Appendix A.2). Overall, we observed that Vintix is not yet capable to handle significantly new tasks. Figure 7 presents one successful rollout and one failure case for each domain (for detailed task-level results, refer to Appendix F). On the *Door-Unlock* task from Meta-World, Vintix achieved 47% of the mean expert-normalized score, while on the *Door-Open-Inward* task from Bi-DexHands, it consistently maintained 31% of demonstrator performance. In failure

scenarios, the model exhibited random-level performance on the *Bin-Picking* task and performed below the random baseline on the *Hand-Kettle* task.

This observation aligns with our intuition and existing research in in-context RL, where even in toy task settings, such as those studied by Laskin et al. (2022); Sinii et al. (2024); Zisman et al. (2024a), authors employed significantly larger task sets in their experiments. Nevertheless, we believe this result is still promising as Vintix manages to demonstrate inference-time improvement through trial and error on certain tasks, despite being trained on only 87 distinct tasks.

## 4. Related Work

**In-Context Learning.** In-Context learning is often used to describe the capacity of large language models to adapt to new tasks after the training phase (Brown et al., 2020;

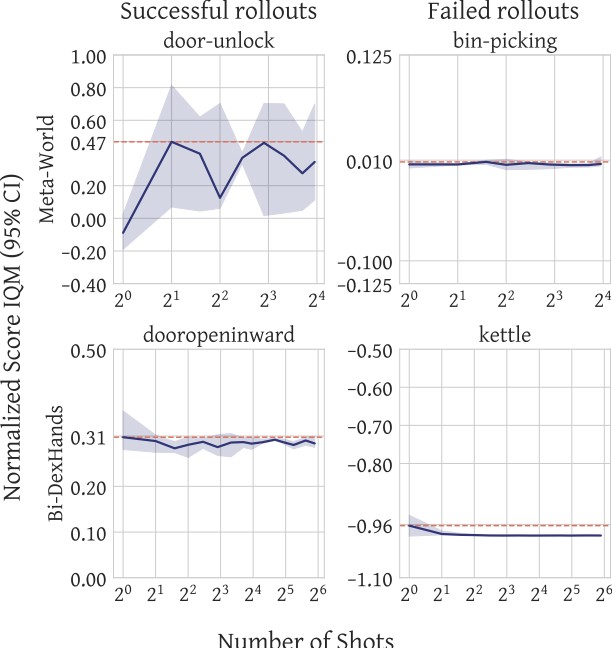

*Figure 7.* Inference-time performance on new tasks. One successful rollout and one failure case are reported for both Meta-World and Bi-DexHands. Inference is performed without an initial context in a task-agnostic manner.

Liu et al., 2021). In essence, in-context learning refers to an approach where the algorithm is provided with a set of demonstrations at test time, enabling it to infer task-related information (Min et al., 2022). In contrast, the in-weights paradigm typically relies on fine-tuning the model on downstream tasks (Finn et al., 2017; Wang et al., 2024; Ying et al., 2024). Compared to in-weights learning, in-context learning is gradient-free at deployment, which theoretically allows for a significant reduction in computational costs and facilitates the development of foundational models as a service, applicable to a broad range of real-world tasks (Sun et al., 2022; Dong et al., 2024). In our work, we use the many-shot in-context learning setup (Agarwal et al., 2024) with a large context window to the reinforcement learning framework.

**Offline Memory-Based Meta-RL.** Meta-reinforcement learning (Meta-RL) focuses on enabling agents to adapt to new tasks, environments, or dynamics through interaction experience. Numerous diverse approaches exist within the field of Meta-RL (Beck et al., 2024). At a high level, Meta-RL algorithms can be broadly categorized into two main segments: those explicitly conditioned on a task representation (Espeholt et al., 2018; Rakelly et al., 2019; Zhao et al., 2020; Sodhani et al., 2021) and those that infer task dynamics and reward functions from past experience often referred to as "In-Context". Implicit memory-based Meta-

RL can itself be divided into two major branches: a set of approaches inheriting from RL$^2$ (Duan et al., 2016) which encodes task-related information using the RNN's hidden state and directly leverages RL off-policy updates, including more recent transformer-based variants like AMAGO-$\{1, 2\}$ (Grigsby et al., 2024a;b) and RELIC (Elawady et al., 2024). Another perspective on in-context reinforcement learning formalizes the training of a Meta-RL agent as an imitation learning problem. This can involve cloning optimal actions, as seen in methods like DPT (Lee et al., 2023), leveraging demonstrator's trajectories, as in ICRT (Fu et al., 2024), or utilizing the entire RL algorithm's learning history or its approximations derived via noise distillation (Zisman et al., 2024a). Examples of this data-centric approach include AD and its derivatives (Laskin et al., 2022; Kirsch et al., 2023; Sinii et al., 2024; Nikulin et al., 2024). Our algorithm is most closely aligned with the last described category and represents a multi-domain AD trained on trajectories obtained through noise distillation.

**Multi-Task Learning.** Multi-Task Learning (MTL) can be formalized as a paradigm of joint multi-task optimization, aiming to maximize positive knowledge transfer (synergy) between tasks while minimizing detrimental task interference. Numerous studies explore Multi-Task Learning (MTL) beyond the naive minimization of the sum of individual task losses, a method commonly referred to as unitary scalarization. Significant efforts have been made in the fields of massively multilingual translation, where each language pair is treated as a separate task (McCann et al., 2018; Radford et al., 2019; Wang et al., 2020; Li & Gong, 2021), as well as in reinforcement learning (Espeholt et al., 2018; Hessel et al., 2018; Sodhani et al., 2021; Kumar et al., 2023) and robotics (Wulfmeier et al., 2020; Wang et al., 2024). The most common solutions include various types of gradient surgeries to minimize negative interactions between different tasks (Wang et al., 2020; Yu et al., 2020), adaptive tuning of task weights (Sener & Koltun, 2019; Li & Gong, 2021), scaling of model size, and temperature tuning (Shaham et al., 2023). It is also worth noting that several studies argue there is no substantial evidence that specialized multi-task optimizers consistently outperform unitary scalarization, while also introducing significant complexity and computational overhead (Xin et al., 2022; Kurin et al., 2023). It is important to note that the aforementioned works on Multi-Task RL rely on various forms of explicit task conditioning, while our approach follows a task-agnostic paradigm that implicitly infers task-related information from interaction experience.

**Generalist Agents and Large Action Models.** Although most experiments in the field of meta-reinforcement learning (Meta-RL) are typically confined to a single domain of tasks (Duan et al., 2016; Anand et al., 2021; Laskin et al., 2022),

generalist agents aim to perform cross-domain training, often integrating multiple data modalities (Gallouédec et al., 2024; Reed et al., 2022). Primary advances in this area of research have been made in the field of robotic locomotion and manipulation, where researchers aim to develop generalizable policies and facilitate cross-domain knowledge transfer. This approach seeks to reduce the computational complexity of training policies for robotic applications. The renaissance of generalist robot policies has been notably driven by the availability of large open-source multi-domain robotic datasets like Open X-Embodiment (Collaboration et al., 2024). A variety of models build upon the foundations established by the Open X datasets, including RT-X (Collaboration et al., 2024), RT-1 (Brohan et al., 2023), Octo (Team et al., 2024), OpenVLA (Kim et al., 2024), DynaMo (Cui et al., 2024). Another category of models relies primarily on training data derived from simulated environments, with notable examples including JAT (Gallouédec et al., 2024), Gato (Reed et al., 2022) and Baku (Haldar et al., 2024). Other closely related work is by Sridhar et al. (2024), where authors propose a retrieval-based algorithm for in-context imitation in the presence of demos for new tasks. Although our work closely aligns with the field of generalist agents and works mentioned, our primary focus is on inference-time learning through trial and error driven by the data-centric approach.

**Learned Optimizers.**  In contrast to traditional optimizers that follow hand-crafted update rules, learned optimizers employ a parameterized update rule that is meta-trained to optimize various objective functions (Li & Malik, 2016; Andrychowicz et al., 2016; Metz et al., 2020; Almeida et al., 2021). Thus, learned optimization can be seen as an alternative perspective on meta-learning (learning-to-learn), with recent approaches scaling to a wide range of tasks and requiring thousands of TPU-months for training (Metz et al., 2022). However, due to the non-stationarity and high stochasticity of the temporal difference (TD) objective, learned optimizers fail in reinforcement learning setting (Metz et al., 2022). To address these challenges, Optim4RL introduces RL-specific inductive bias (Lan et al., 2024), while OPEN enhances exploration by leveraging learnable stochasticity (Goldie et al., 2024). Both works can be considered optimization-centric approaches to Meta-RL, whereas we adopt a context-based approach.

**Sequence Modeling in RL.**  With the increasing use of Transformers for modeling sequential data, several concurrent works (Chen et al., 2021; Janner et al., 2021) reformulated the Markov Decision Process as a causal sequence modeling problem. Chen et al. (2021) focused on reward conditioning and treated each component of the MDP as a separate token, while Janner et al. (2021) applied beam search over discretized states, actions, and rewards.

Subsequent research has expanded this direction by encouraging such models to maximize returns (Zhuang et al., 2024), adapting Decision Transformers to online learning settings (Zheng et al., 2022), and replacing the Transformer backbone with architectures that scale more efficiently with input length, such as Mamba (Huang et al., 2024).

Another line of work has aimed to scale this modeling paradigm to multi-domain and multi-modal environments (Reed et al., 2022; Gallouédec et al., 2024), or to leverage the in-context learning capabilities of Transformers for Meta-RL (Laskin et al., 2022; Lee et al., 2023). This latter direction is most closely related to Vintix, which is a memory-based, cross-domain Meta-RL method.

## 5. Conclusion and Future Work

The development of generalist reinforcement learning agents that adapt across domains remains a critical challenge, as traditional online RL methods—despite their success in narrow settings—face scalability limitations due to their reliance on environment-specific, interactive training (Dulac-Arnold et al., 2019; Levine et al., 2020). While recent advances in offline RL and generative action models have expanded the potential for data-driven learning, these approaches often prioritize expert demonstrations or language conditioning over reward-centric adaptation (Reed et al., 2022; Gallouédec et al., 2024; Haldar et al., 2024; Collaboration et al., 2024; Schmied et al., 2024; Sridhar et al., 2024). In this work, we explored an alternative paradigm rooted in In-Context Reinforcement Learning (ICRL), building on Algorithm Distillation (Laskin et al., 2022) to create agents that learn adaptive behaviors by a straightforward next-action prediction of learning histories or their proxies.

Our proposed approach and model, Vintix, demonstrates that ICRL can extend beyond prior single-domain, grid-based benchmarks. By introducing Continuous Noise Distillation, we ease the data collection process inherent to Algorithm Distillation and release a suite of datasets and tools for 87 tasks across four domains, which we hope would be helpful in community efforts toward scalable, cross-domain action models capable of in-context reinforcement learning. Empirically, we show that Vintix exhibits self-correction on training tasks and adapt to moderate controlled parametric variations without requiring gradient updates at inference time. These results, though preliminary and confined to structured settings, suggest that reward-guided ICRL provides a viable pathway for agents to autonomously refine their policies in response to environmental feedback.

While we believe the obtained results are promising, there is a large room for improvement as challenges remain in increasing the number of domains, developing more task-agnostic architectures that would allow not only for robust

self-correction but vital generalization to unseen tasks and self-improvement beyond demonstrators' performance. We hope this work encourages further investigation into data-centric, reward-driven frameworks for cross-domain RL agentic systems.

## Acknowledgements

This work was supported by the Ministry of Economic Development of the RF (code 25-139-66879-1-0003).

## Impact Statement

This proposed model is strictly limited to the simulated environments used for evaluation due to its architectural limitations, and therefore does not present any safety concerns. However, in case one would like to finetune it for real-world purposes, it does not guarantee any sort of safe behavior and cautions must be made on the part of the user.

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

# A. Dataset Details

## A.1. General Information

**MuJoCo.** MuJoCo (Todorov et al., 2012) is a physics engine designed for multi-joint continuous control. For our purposes, we selected 11 classic continuous control environments from OpenAI Gym (Brockman et al., 2016) and Gymnasium (Towers et al., 2024).

**Meta-World.** Meta-World (Yu et al., 2021) is an open-source simulated benchmark for meta-reinforcement learning and multitask learning consisting of 50 distinct robotic manipulation tasks. It is important to note that, similar to JAT (Gallouédec et al., 2024), we limit the episode length to 100 transitions to reduce dataset size and increase the number of episodes that fit within the model's context. This truncation is feasible, as such a horizon is often sufficient to solve the task. However, this modification complicates direct performance comparisons with models such as AMAGO-2 (Grigsby et al., 2024b), which utilize full 500-step rollouts and reports total returns over three consecutive episodes.

**Bi-DexHands.** Bi-DexHands (Chen et al., 2022) is the first suite of bi-manual manipulation environments designed for practitioners in common RL, MARL, offline RL, multi-task RL, and Meta-RL. It provides 20 tasks that feature complex, high-dimensional continuous action and observation spaces. Some task subgroups share state and action spaces, making this benchmark applicable to the Meta-RL framework.

**Industrial-Benchmark.** Industrial-Benchmark (Hein et al., 2017) is a suite of synthetic continuous control problems designed to model various aspects that are crucial in industrial applications, such as the optimization and control of gas and wind turbines. The dynamic behavior of the environment, as well as its stochasticity, can be controlled through the setpoint parameter $p$. By varying this parameter from 0 to 100 in increments of 5, we generated 21 tasks that share a common state and action structure. Classic reward function was selected and subsequently down-scaled by a factor of 100.

## A.2. Train vs. Test Tasks Split

To validate the inference-time optimization capability of our model, we divided the overall set of 102 tasks into two disjoint subsets. The validation subset was excluded from the training dataset. Below, we provide details of the split for each domain.

### A.2.1. MuJoCo

Validation tasks for the MuJoCo domain were created by modifying the physical parameters of the environments using the provided XML API. For each embodiment, the default viscosity parameter was adjusted from 0 to 0.05, and 0.1. Additionally, the gravity parameter was varied by ±10 percent.

### A.2.2. Meta-World

The standard ML45 split was selected, with 45 tasks assigned to the training set and 5 tasks reserved for validation: *bin-picking*, *box-close*, *door-lock*, *door-unlock*, and *hand-insert*.

### A.2.3. Bi-DexHands

We adopted the ML20 benchmark setting proposed by the original authors (Chen et al., 2022), in which 15 tasks are assigned to the training set, while 5 tasks are reserved for validation, including: *door-close-outward*, *door-open-inward*, *door-open-outward*, *hand-kettle*, and *hand-over*. It is important to note that the *hand-over* task does not share the same state-action space dimensionality with any tasks in the training set, making it incompatible with the current encoder architecture. As a result, we report a performance of 0 (random) for this task.

### A.2.4. Industrial-Benchmark

For this domain, a global split based on the setpoint parameter was made, with setpoints ranging from 0 to 75 assigned to the training set, and setpoints from 80 to 100 assigned to the validation set.

### A.3. Epsilon Decay Functions

The epsilon decay function defines how the noise proportion $\epsilon$ depends on the transition number $n_s$ within trajectory. The primary purpose of utilizing such function is to ensure in smooth increase in the rewards through generated trajectories. A linear decay function may not yield this behavior across all environments, as some tasks exhibit rewards that increase rapidly either at the beginning or the end of trajectories when linear function is utilized. To address this variability, we employ the following generalized decay function:

$$\epsilon(n_s) = \begin{cases} \sqrt[p]{1 - (n_s/((1-f)N_s))^p} & n_s <= (1-f)N_s \\ 0 & n_s > (1-f)N_s \end{cases}$$

where $N_s$ represents the maximum number of transitions in a trajectory, $f$ the fraction of the trajectory with zero noise, and $p$ is a parameter that controls the curvature of the decay function. By adjusting $p$, it is possible to modulate the smoothness of the reward progression along the trajectory in order to better suit different tasks. Specifically, when rewards increase sharply at the beginning of the trajectory, setting $p > 1$ flattens the epsilon curve at the start and steepens it at the end. Conversely, when rewards increase in a sharp manner towards the end of the trajectory, choosing $0 < p < 1$ results in the epsilon curve being steepened at the start and flattened at the end.

## B. Demonstrators

### B.1. Training

**MuJoCo.**   The dataset provided by JAT (Gallouédec et al., 2024) consists of demonstrations from expert RL agents on MuJoCo (Todorov et al., 2012) tasks. It was used to train demonstrators through the imitation learning paradigm. The resulting models achieve performance similar to that of the original RL agents.

**Meta-World.**   For the Meta-World (Yu et al., 2021) benchmark, Gallouédec et al. (2024) open-sourced trained agents for each task in the benchmark. These models served as demonstrators; however, a detailed analysis revealed that certain agents exhibited suboptimal performance. Specifically, models trained on the *disassemble*, *coffee-pull*, *coffee-push*, *soccer*, *push-back*, *peg-insert-side*, and *pick-out-of-hole* tasks were either too noisy or performed unsatisfactorily. As a result, we retrained agents for these tasks.

JAT (Gallouédec et al., 2024) also provided open-source training scripts, which we utilized for hyperparameter tuning and retraining new agents. This process led to improved performance for the selected tasks. However, many demonstrators across other tasks still exhibit noisy performance.

**Bi-DexHands.**   Demonstrators were trained using the provided PPO (Schulman et al., 2017) implementation. To enhance convergence, the number of parallel environments in IsaacGym (Liang et al., 2018) was increased from 128 to 2048. In certain environments, such as the *Re-Orientation* and *Swing-Cup* tasks, we were able to significantly surpass the expert performance reported in the original Bi-DexHands paper (Chen et al., 2022). However, even after extensive training for over 1.5 billion timesteps, some policies continued to exhibit stochastic performance.

**Industrial-Benchmark.**   Demonstrators were trained using the PPO implementation from Stable-Baselines3 (Raffin et al., 2021). For all tasks, PPO was trained with advantage normalization, a KL-divergence limit of 0.2, 2500 environment steps, and a batch size of 50. All agents were trained for 1 million timesteps. Similarly to original Industrial Benchmark setup (Hein et al., 2017), discount factor was set to 0.97. To ensure better score comparability, we limited the episode length to 250 interactions. PPO agents were trained using both classic and delta rewards; however, we observed that agents trained on delta rewards achieved higher classic reward scores compared to those trained directly on scaled classic returns. Consequently, for our experiments, we utilize demonstrators trained with delta rewards.

# C. Additional Experiments

### C.1. Is Algorithm Distillation More Effective Than Expert Distillation?

To assess the benefits of AD-style training on noise-distilled trajectories over simple Expert Distillation (ED), we conducted an experiment using a model with the same architecture as Vintix—including the transformer backbone, encoders, and loss function—but trained exclusively on expert-level demonstrations. We then evaluated both the ED and AD models using the cold-start inference procedure.

Figure 8 shows that ED underperforms relative to AD in terms of asymptotic performance across both the MuJoCo and Meta-World domains. Specifically, ED converges to an average expert-normalized score of 0.8, whereas AD reaches 0.97 on MuJoCo and 0.95 on Meta-World.

These results highlight the critical role of policy-improvement structure in the dataset for enabling self-correcting behavior and achieving superior performance. Notably, while ED's performance remains relatively stable across different shot counts in MuJoCo, it shows a positive trend in Meta-World as the number of shots increases.

### C.2. Does Vintix Performs In-Context Reinforcement Learning?

To investigate whether Vintix engages in reinforcement learning during inference, we trained a variant of the model without access to reward signals. We then compared its performance to the original Vintix model (trained with rewards) using the cold-start inference procedure on training tasks from both domains.

Evaluation results (Figure 8) demonstrate that removing the reward signal significantly impairs performance. The reward-masked AD model performs worse asymptotically and exhibits slower convergence on the Meta-World domain.

These findings suggest that reward feedback is essential for effective self-improvement during inference. This supports the hypothesis that supervised training on a dataset containing policy improvement mechanisms enhances the model's in-context reinforcement learning capabilities.

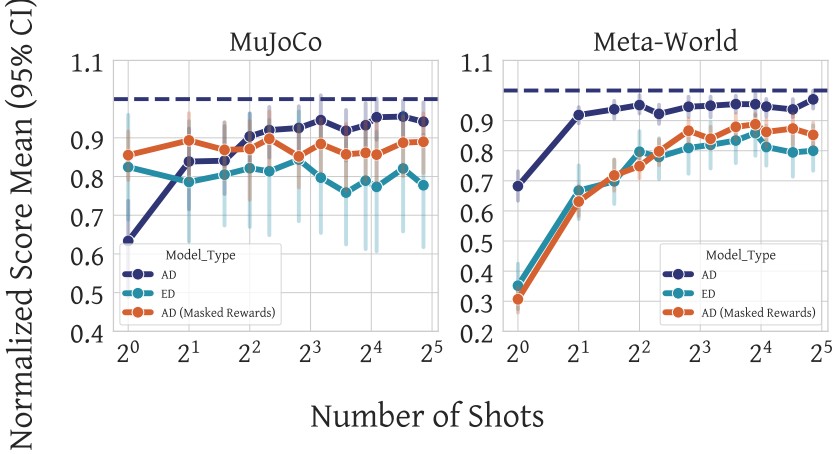

*Figure 8.* Performance of AD, ED and AD without rewards on MuJoCo and Meta-World domains.

# D. Hyperparameters

| Hyperparameter | Value |
|---|---|
| Learning Rate | 0.0003 |
| Optimizer | Adam |
| Beta 1 | 0.9 |
| Beta 2 | 0.99 |
| Batch Size | 64 |
| Gradient Accumulation Steps | 2 |
| Transformer Layers | 20 |
| Transformer Heads | 16 |
| Context Length | 8192 |
| Transformer Hidden Dim | 1024 |
| FF Hidden Size | 4096 |
| MLP Type | GptNeoxMLP |
| Normalization Type | LayerNorm |
| Training Precision | bf16 |
| Parameters | 332100768 |

*Table 2.* Hyperparameters used in training.

## E. Task-Level Dataset Visualization

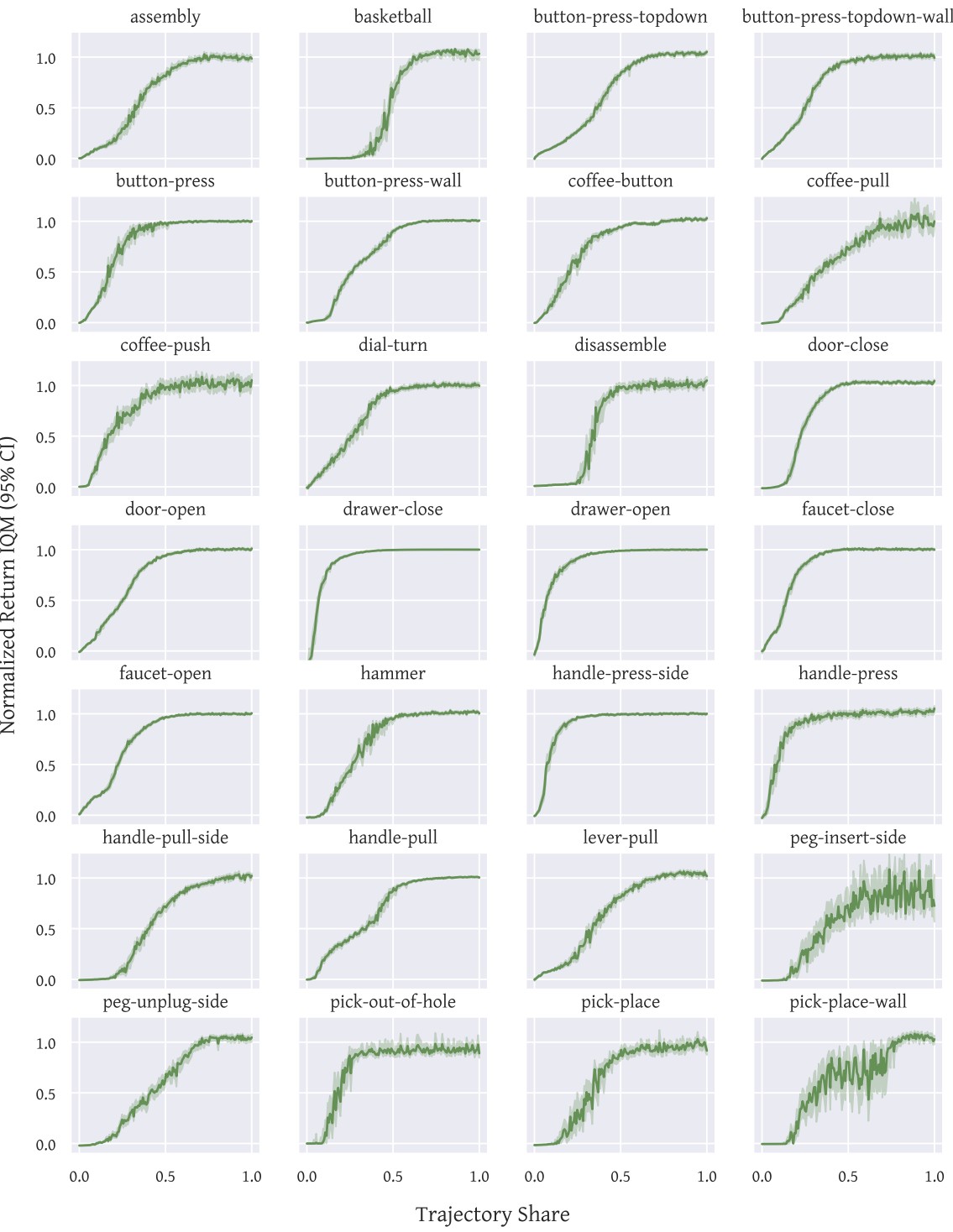

*Figure 9.* IQM aggregated noise-distilled trajectories for Meta-World domain (Part 1 of 2)

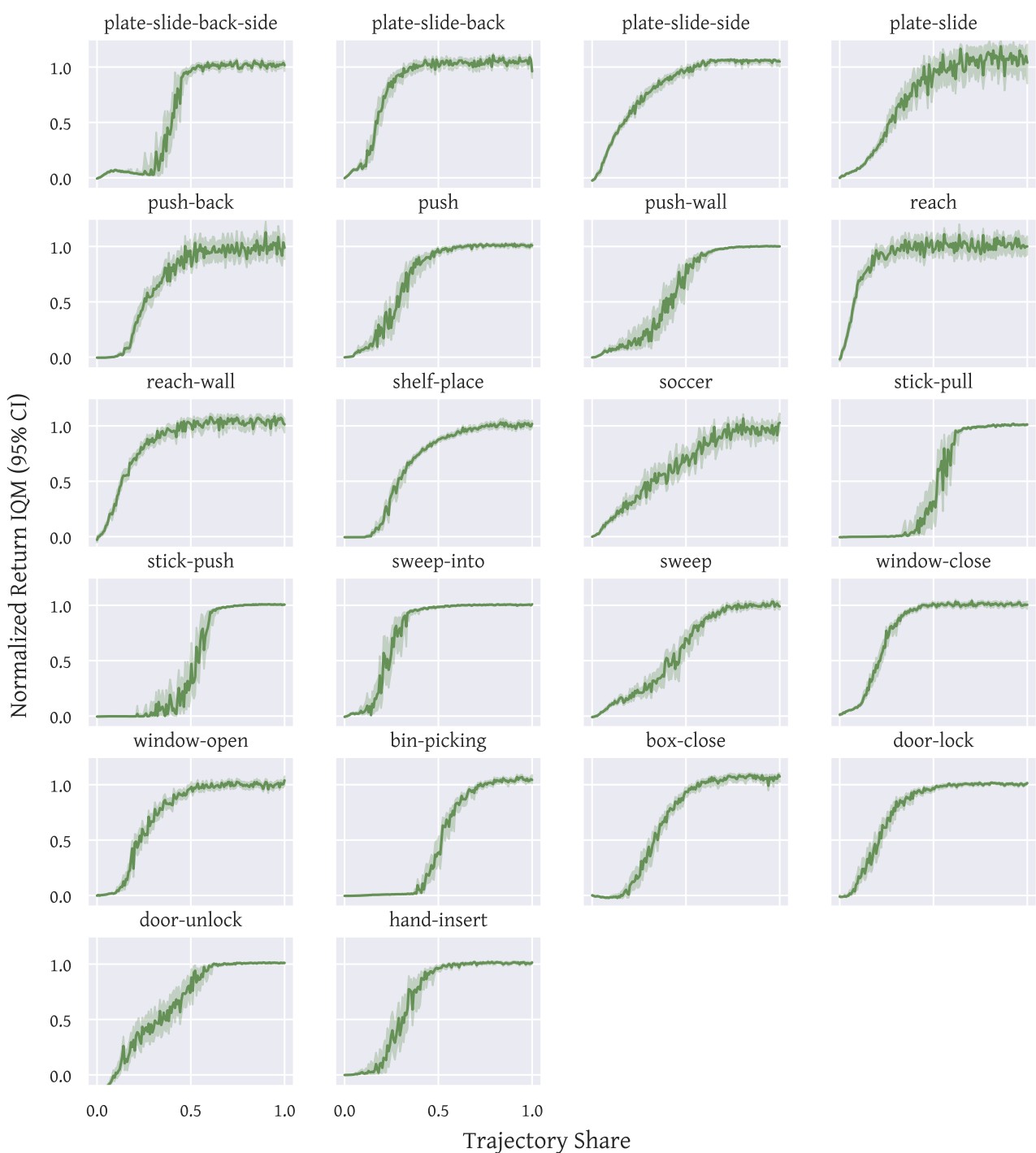

*Figure 10.* IQM aggregated noise-distilled trajectories for Meta-World domain (Part 2 of 2)

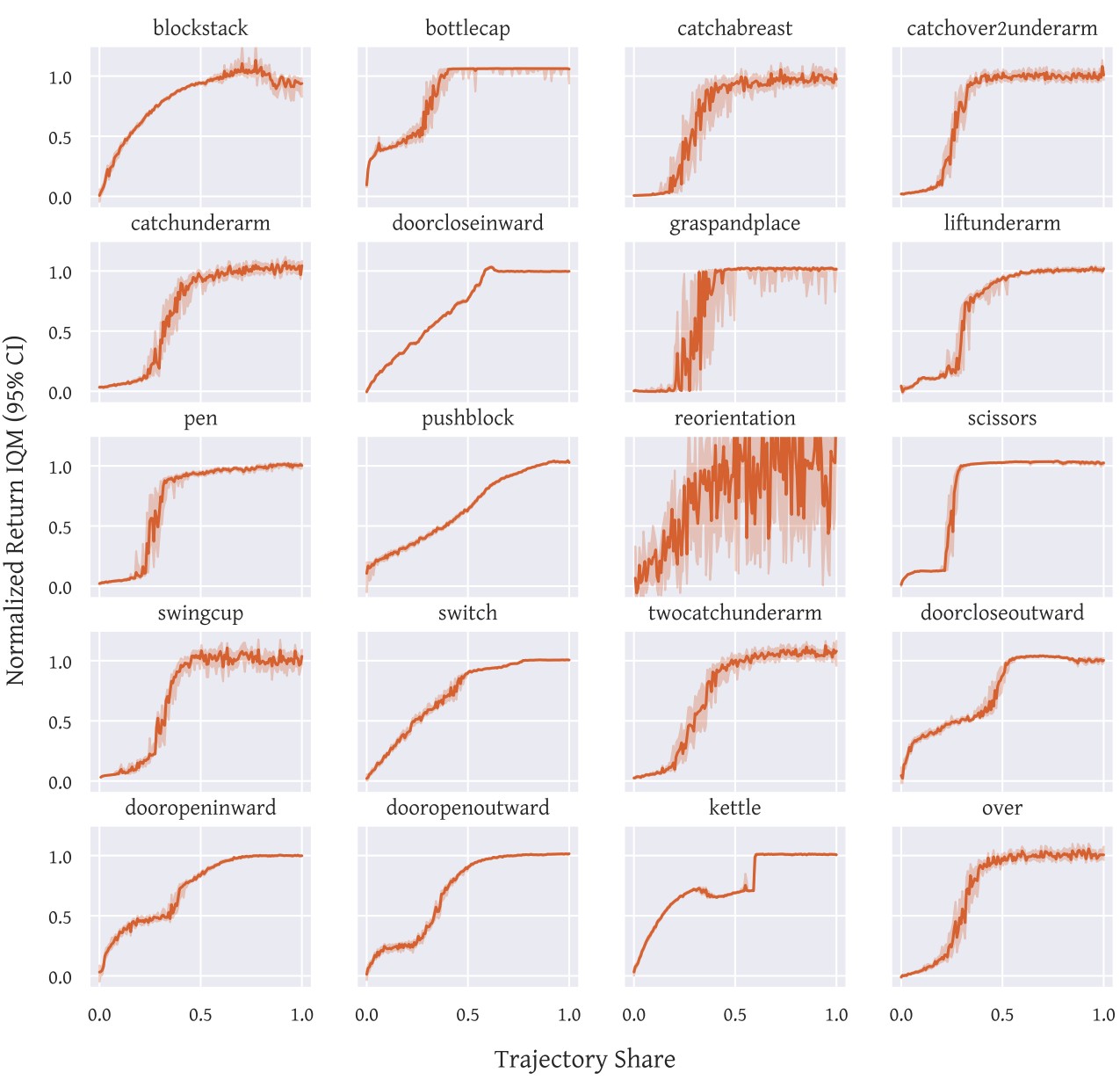

*Figure 11.* IQM aggregated noise-distilled trajectories for Bi-DexHands domain

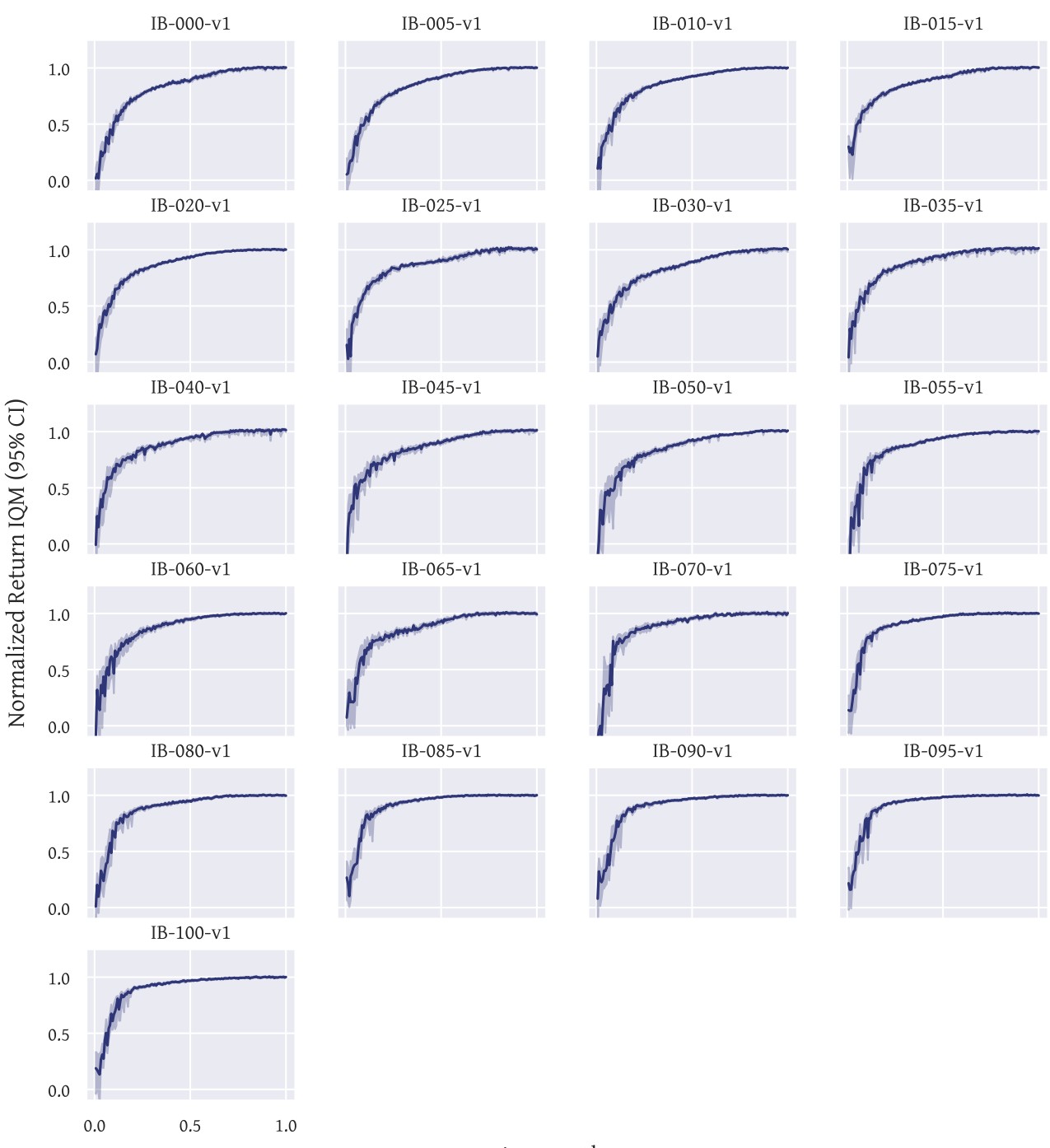

*Figure 12.* IQM aggregated noise-distilled trajectories for Industrial-Benchmark domain

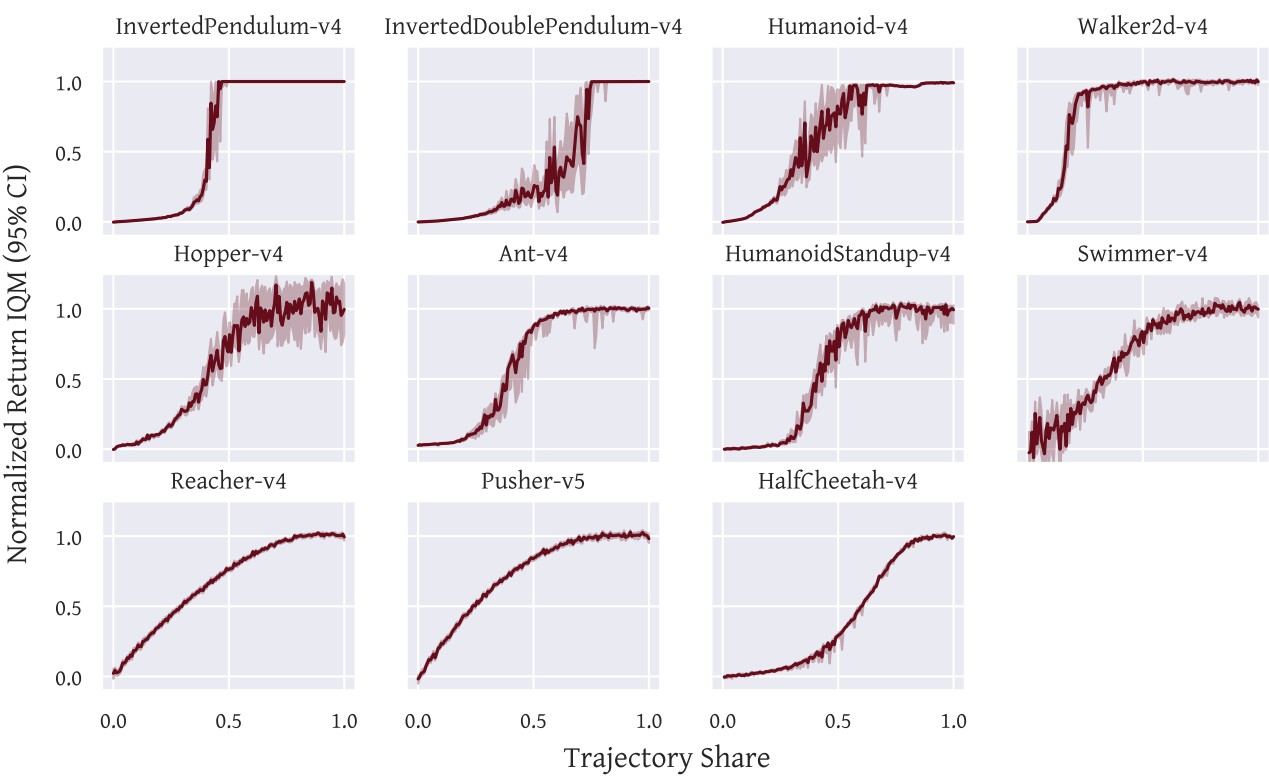

*Figure 13.* IQM aggregated noise-distilled trajectories for MuJoCo domain

## F. Inference Time Performance Graphs

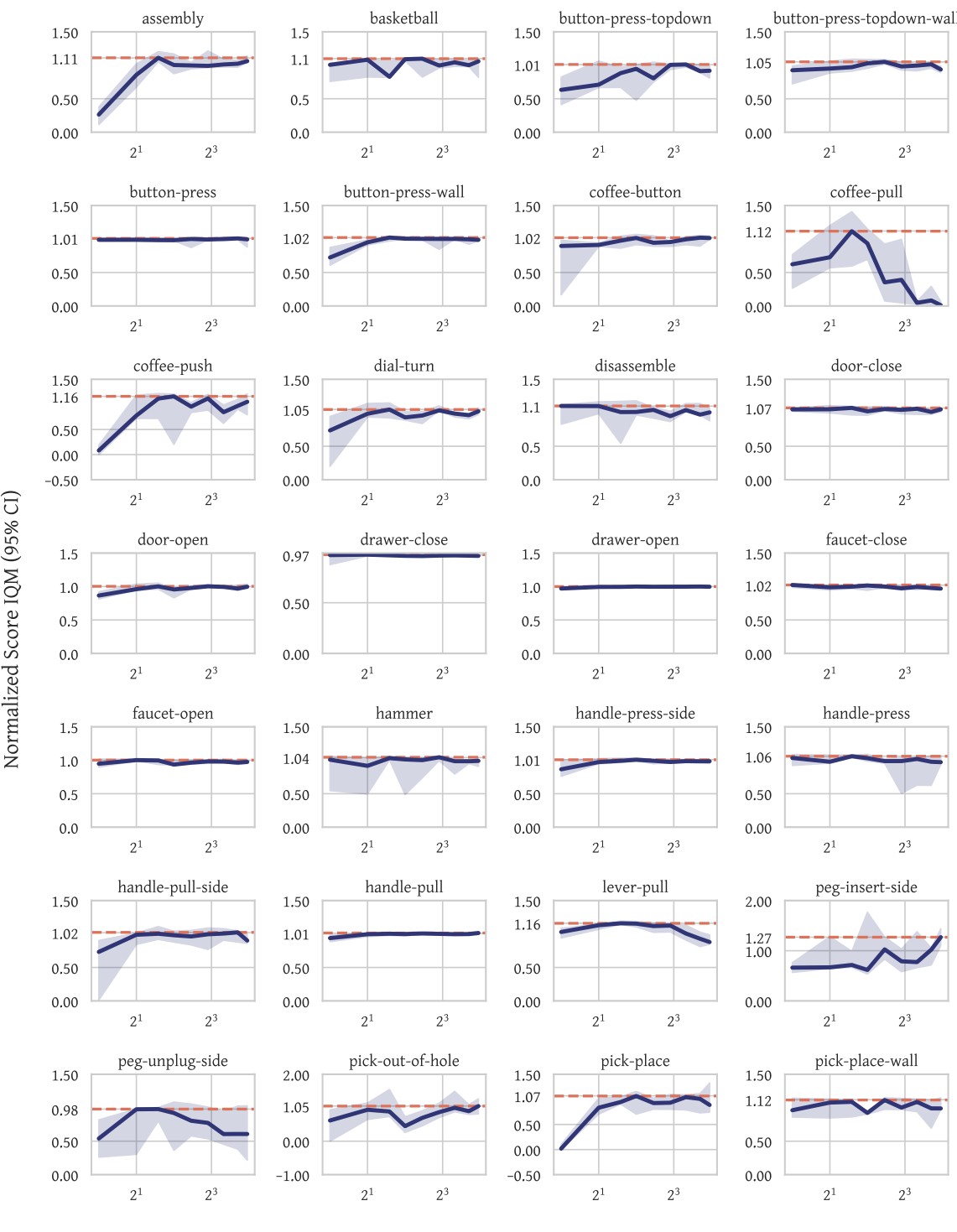

*Figure 14.* Inference performance for Meta-World domain (Part 1 of 2)

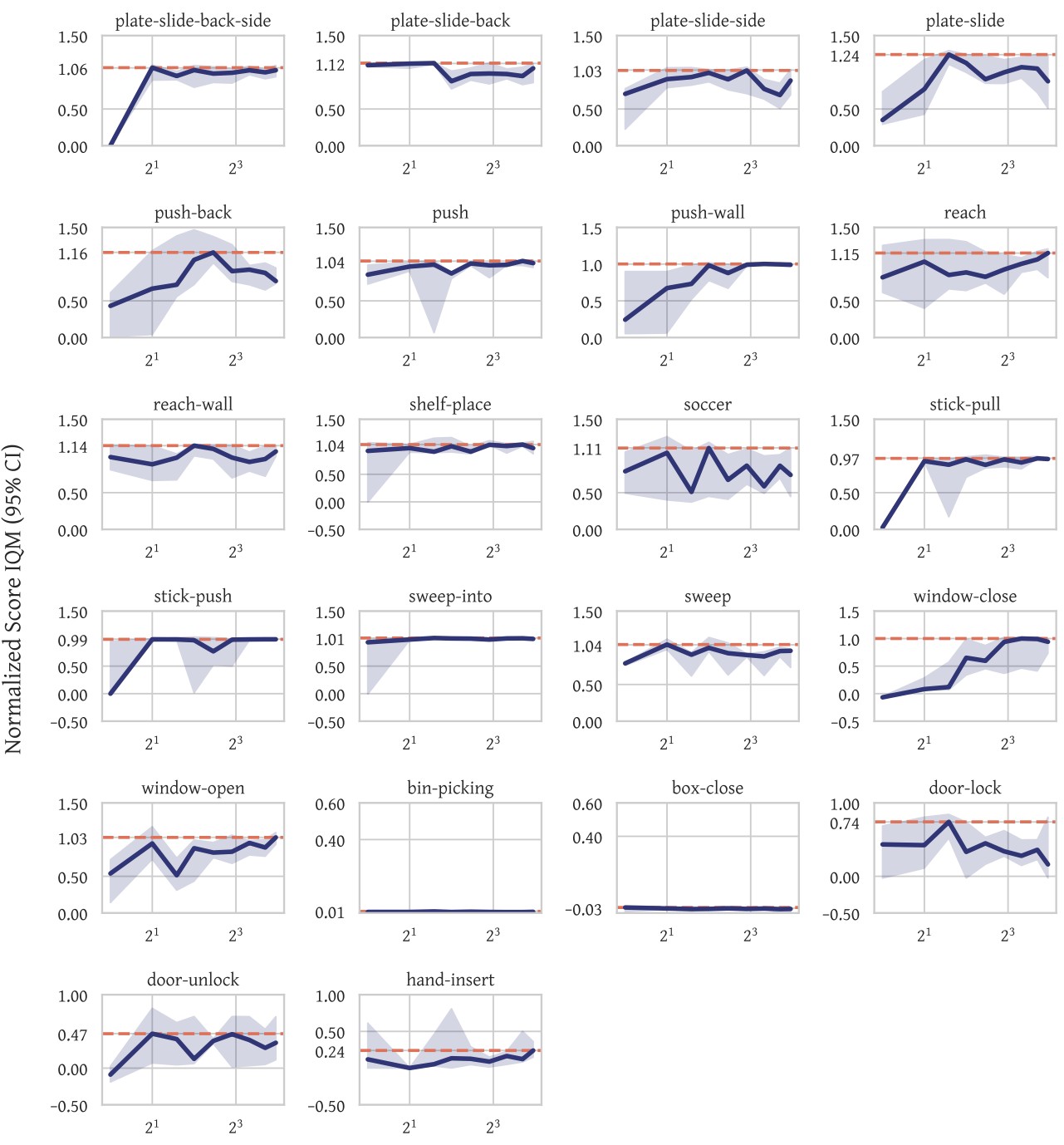

*Figure 15.* Inference performance for Meta-World domain (Part 2 of 2)

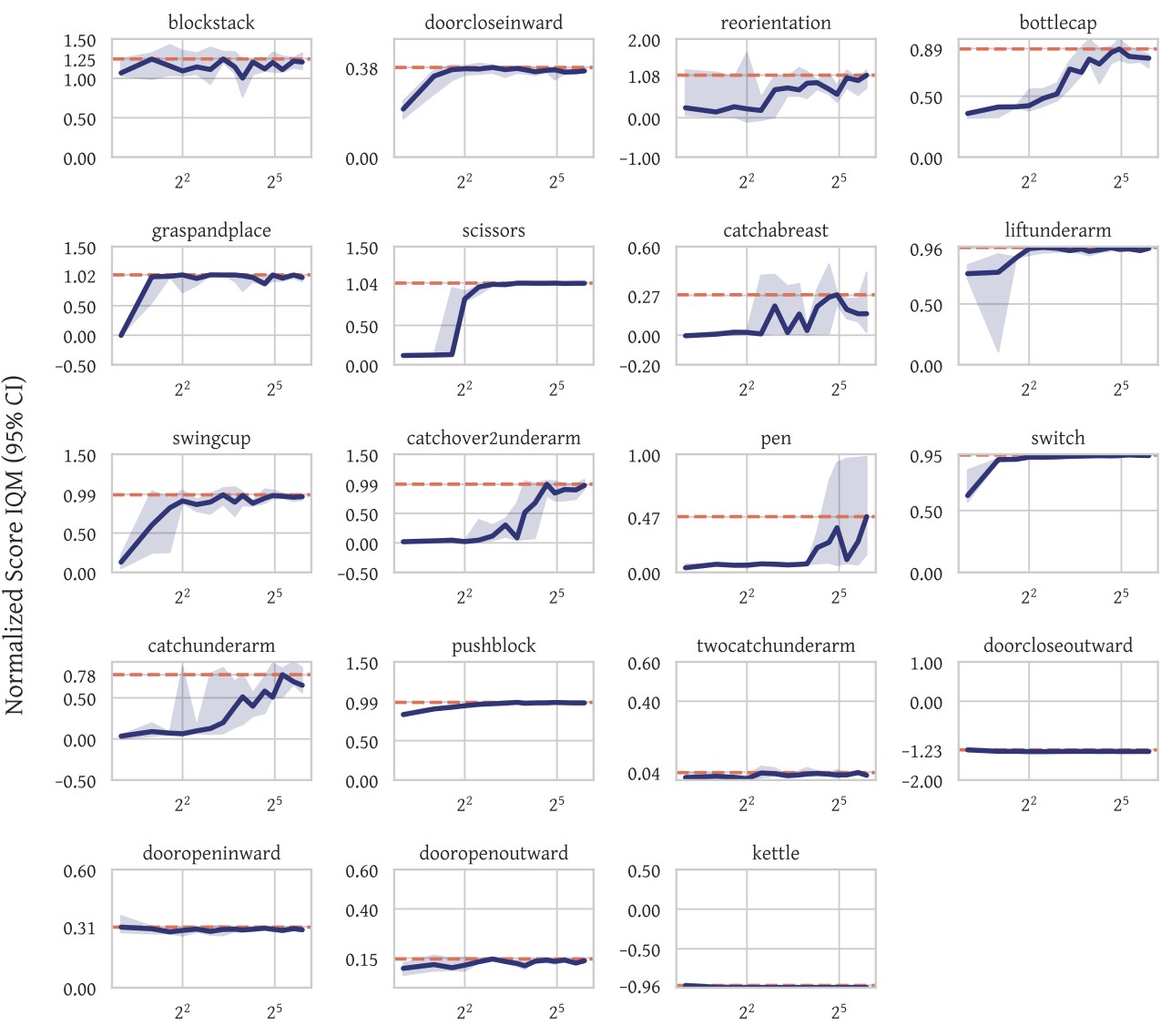

*Figure 16.* Inference performance for Bi-DexHands domain

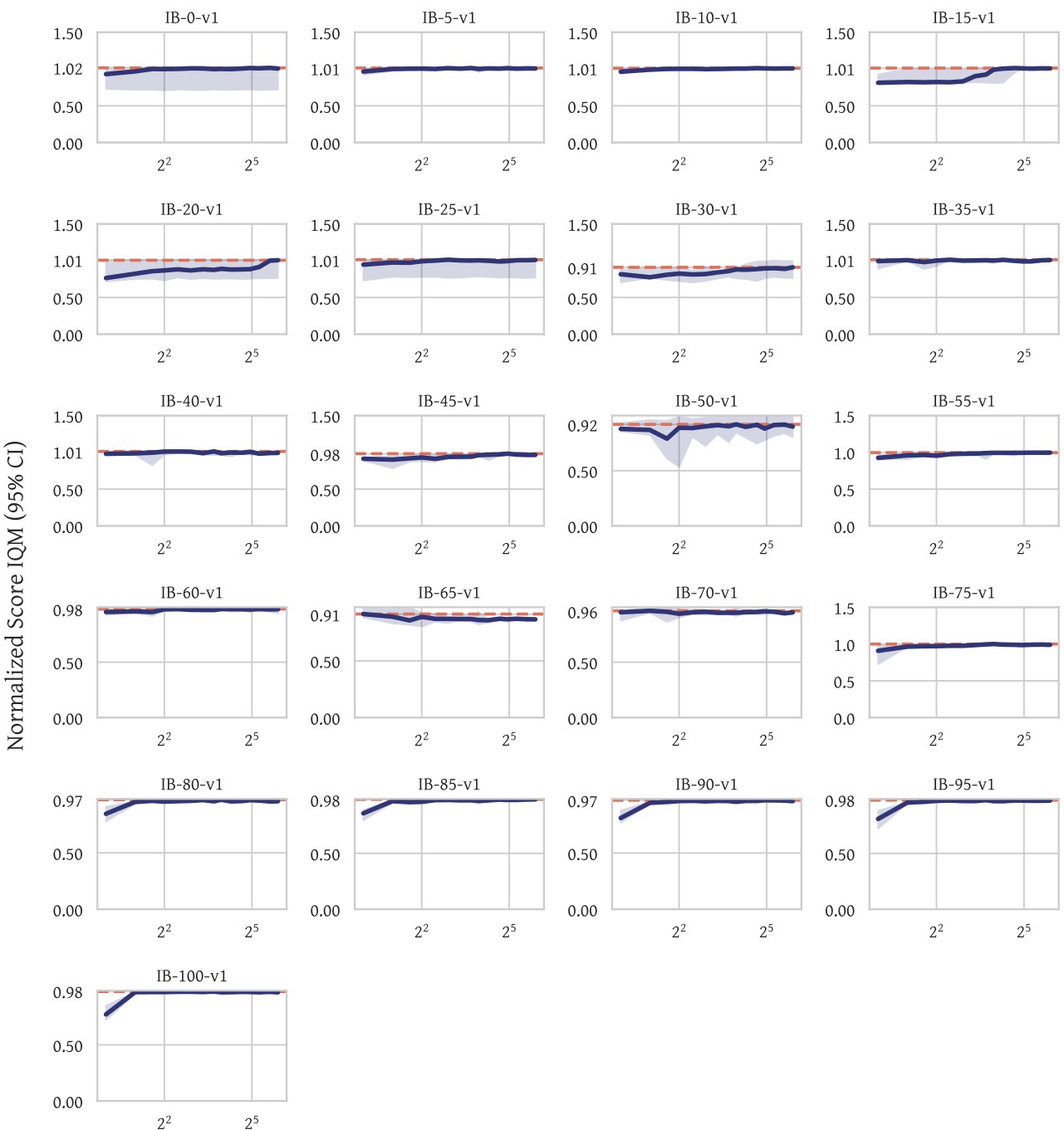

*Figure 17.* Inference performance for Industrial-Benchmark domain

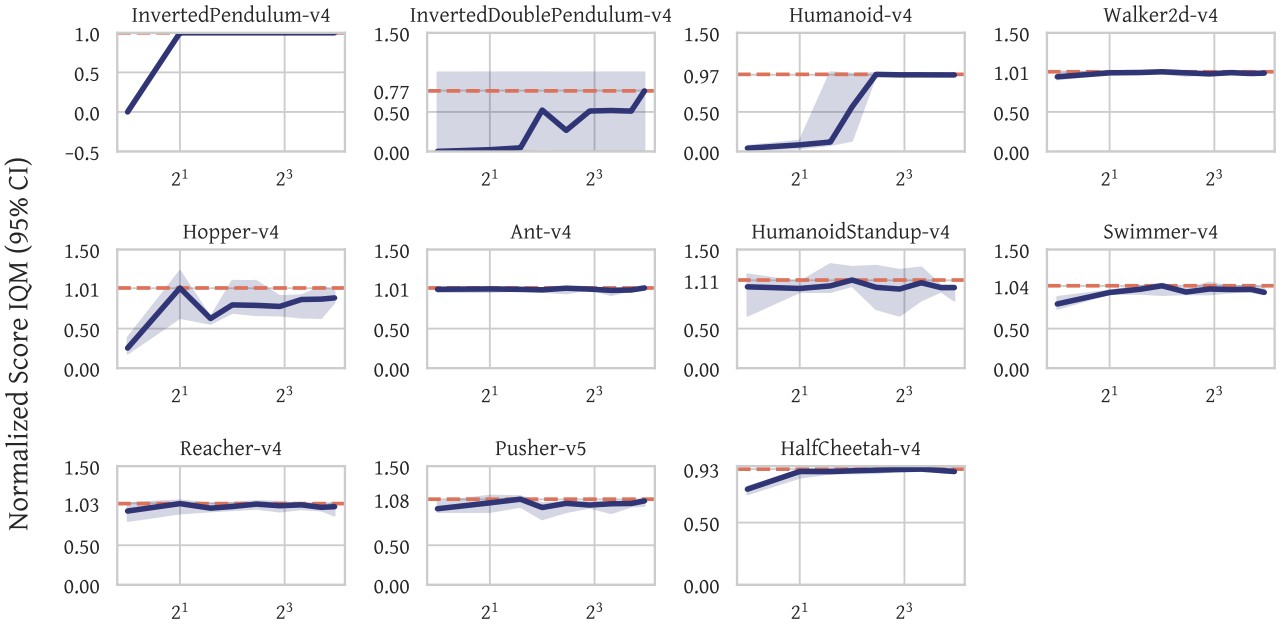

*Figure 18.* Inference performance for MuJoCo domain

## G. Dataset Size and Metadata

| Task | Trajectory Length | Number of trajectories | Mean Episode Length | State space shape | Action space shape | Reward Scaling |
|---|---|---|---|---|---|---|
| Hand-Block-Stack | 250000 | 10 | 248.6 | (428, ) | (52, ) | 1.0 |
| Hand-Bottle-Cap | 135000 | 10 | 123.7 | (420, ) | (52, ) | 1.0 |
| Hand-Catch-Abreast | 165000 | 10 | 98.5 | (422, ) | (52, ) | 1.0 |
| Hand-Catch-Over-2-Underarm | 100000 | 10 | 55.6 | (422, ) | (52, ) | 1.0 |
| Hand-Catch-Underarm | 100000 | 10 | 64.9 | (422, ) | (52, ) | 1.0 |
| Hand-Door-Close-Inward | 100000 | 10 | 23.8 | (417, ) | (52, ) | 1.0 |
| Hand-Grasp-And-Place | 500000 | 8 | 332.6 | (425, ) | (52, ) | 1.0 |
| Hand-Lift-Underarm | 500000 | 10 | 455.0 | (417, ) | (52, ) | 1.0 |
| Hand-Pen | 135000 | 10 | 120.5 | (417, ) | (52, ) | 1.0 |
| Hand-Push-Block | 135000 | 10 | 123.2 | (428, ) | (52, ) | 1.0 |
| Hand-Reorientation | 600000 | 7 | 463.9 | (422, ) | (40, ) | 1.0 |
| Hand-Scissors | 175000 | 10 | 149.0 | (417, ) | (52, ) | 1.0 |
| Hand-Swing-Cup | 320000 | 10 | 299.0 | (417, ) | (52, ) | 1.0 |
| Hand-Switch | 135000 | 10 | 124.0 | (417, ) | (52, ) | 1.0 |
| Hand-Two-Catch-Underarm | 100000 | 10 | 65.1 | (446, ) | (52, ) | 1.0 |
| Hand-Door-Close-Outward | 265000 | 10 | 249.0 | (417, ) | (52, ) | 1.0 |
| Hand-Door-Open-Inward | 265000 | 10 | 249.0 | (417, ) | (52, ) | 1.0 |
| Hand-Door-Open-Outward | 250000 | 10 | 249.0 | (417, ) | (52, ) | 1.0 |
| Hand-Kettle | 135000 | 10 | 124.0 | (417, ) | (52, ) | 1.0 |
| Hand-Over | 100000 | 10 | 64.4 | (398, ) | (40, ) | 1.0 |

*Table 4.* Bi-DexHands - Detailed information about collected dataset

| Task | Trajectory Length | Number of trajectories | Mean Episode Length | State space shape | Action space shape | Reward Scaling |
|------|------|------|------|------|------|------|
| assembly | 100000 | 15 | 100 | (39,) | (4,) | 1.0 |
| basketball | 100000 | 15 | 100 | (39,) | (4,) | 1.0 |
| bin-picking | 100000 | 5 | 100 | (39,) | (4,) | 1.0 |
| box-close | 100000 | 5 | 100 | (39,) | (4,) | 1.0 |
| button-press | 100000 | 15 | 100 | (39,) | (4,) | 1.0 |
| button-press-topdown | 100000 | 15 | 100 | (39,) | (4,) | 1.0 |
| button-press-topdown-wall | 100000 | 15 | 100 | (39,) | (4,) | 1.0 |
| button-press-wall | 100000 | 15 | 100 | (39,) | (4,) | 1.0 |
| coffee-button | 100000 | 15 | 100 | (39,) | (4,) | 1.0 |
| coffee-pull | 100000 | 15 | 100 | (39,) | (4,) | 1.0 |
| coffee-push | 100000 | 15 | 100 | (39,) | (4,) | 1.0 |
| dial-turn | 100000 | 15 | 100 | (39,) | (4,) | 1.0 |
| disassemble | 100000 | 15 | 100 | (39,) | (4,) | 1.0 |
| door-close | 100000 | 15 | 100 | (39,) | (4,) | 1.0 |
| door-lock | 100000 | 5 | 100 | (39,) | (4,) | 1.0 |
| door-open | 100000 | 15 | 100 | (39,) | (4,) | 1.0 |
| door-unlock | 100000 | 5 | 100 | (39,) | (4,) | 1.0 |
| drawer-close | 100000 | 15 | 100 | (39,) | (4,) | 1.0 |
| drawer-open | 100000 | 15 | 100 | (39,) | (4,) | 1.0 |
| faucet-close | 100000 | 15 | 100 | (39,) | (4,) | 1.0 |
| faucet-open | 100000 | 15 | 100 | (39,) | (4,) | 1.0 |
| hammer | 100000 | 15 | 100 | (39,) | (4,) | 1.0 |
| hand-insert | 100000 | 5 | 100 | (39,) | (4,) | 1.0 |
| handle-press | 100000 | 15 | 100 | (39,) | (4,) | 1.0 |
| handle-press-side | 100000 | 15 | 100 | (39,) | (4,) | 1.0 |
| handle-pull | 100000 | 15 | 100 | (39,) | (4,) | 1.0 |
| handle-pull-side | 100000 | 15 | 100 | (39,) | (4,) | 1.0 |
| lever-pull | 100000 | 15 | 100 | (39,) | (4,) | 1.0 |
| peg-insert-side | 100000 | 15 | 100 | (39,) | (4,) | 1.0 |
| peg-unplug-side | 100000 | 15 | 100 | (39,) | (4,) | 1.0 |
| pick-out-of-hole | 100000 | 15 | 100 | (39,) | (4,) | 1.0 |
| pick-place | 100000 | 15 | 100 | (39,) | (4,) | 1.0 |
| pick-place-wall | 100000 | 15 | 100 | (39,) | (4,) | 1.0 |
| plate-slide | 100000 | 15 | 100 | (39,) | (4,) | 1.0 |
| plate-slide-back | 100000 | 15 | 100 | (39,) | (4,) | 1.0 |
| plate-slide-back-side | 100000 | 15 | 100 | (39,) | (4,) | 1.0 |
| plate-slide-side | 100000 | 15 | 100 | (39,) | (4,) | 1.0 |
| push | 100000 | 15 | 100 | (39,) | (4,) | 1.0 |
| push-back | 100000 | 15 | 100 | (39,) | (4,) | 1.0 |
| push-wall | 100000 | 15 | 100 | (39,) | (4,) | 1.0 |
| reach | 100000 | 15 | 100 | (39,) | (4,) | 1.0 |
| reach-wall | 100000 | 15 | 100 | (39,) | (4,) | 1.0 |
| shelf-place | 100000 | 15 | 100 | (39,) | (4,) | 1.0 |
| soccer | 100000 | 15 | 100 | (39,) | (4,) | 1.0 |
| stick-pull | 100000 | 15 | 100 | (39,) | (4,) | 1.0 |
| stick-push | 100000 | 15 | 100 | (39,) | (4,) | 1.0 |
| sweep | 100000 | 15 | 100 | (39,) | (4,) | 1.0 |
| sweep-into | 100000 | 15 | 100 | (39,) | (4,) | 1.0 |
| window-close | 100000 | 15 | 100 | (39,) | (4,) | 1.0 |
| window-open | 100000 | 15 | 100 | (39,) | (4,) | 1.0 |

*Table 3.* Meta-World - Detailed information about collected dataset

| Task | Trajectory Length | Number of trajectories | Mean Episode Length | State space shape | Action space shape | Reward Scaling |
|---|---|---|---|---|---|---|
| Ant-v4 | 1000000 | 10 | 524.1 | (105, ) | (8, ) | 0.1 |
| HalfCheetah-v4 | 1000000 | 5 | 1000 | (17, ) | (6, ) | 0.1 |
| Hopper-v4 | 1000000 | 10 | 227.3 | (11, ) | (3, ) | 0.1 |
| Humanoid-v4 | 1000000 | 15 | 207.7 | (348, ) | (17, ) | 0.1 |
| HumanoidStandup-v4 | 1000000 | 15 | 1000 | (348, ) | (17, ) | 0.01 |
| InvertedDoublePendulum-v4 | 1000000 | 10 | 55.7 | (9, ) | (1, ) | 0.1 |
| InvertedPendulum-v4 | 1000000 | 10 | 68.2 | (4, ) | (1, ) | 0.1 |
| Pusher-v5 | 1000000 | 5 | 100 | (23, ) | (7, ) | 0.1 |
| Reacher-v4 | 1000000 | 5 | 50 | (10, ) | (2, ) | 0.1 |
| Swimmer-v4 | 1000000 | 5 | 1000 | (8, ) | (2, ) | 1.0 |
| Walker2d-v4 | 1000000 | 10 | 350.2 | (17, ) | (6, ) | 0.1 |

*Table 5.* MuJoCo - Detailed information about collected dataset

| Task | Trajectory Length | Number of trajectories | Mean Episode Length | State space shape | Action space shape | Reward Scaling |
|---|---|---|---|---|---|---|
| IB-p-0 | 100000 | 15 | 250 | (6, ) | (3, ) | 1.0 |
| IB-p-5 | 100000 | 15 | 250 | (6, ) | (3, ) | 1.0 |
| IB-p-10 | 100000 | 15 | 250 | (6, ) | (3, ) | 1.0 |
| IB-p-15 | 100000 | 15 | 250 | (6, ) | (3, ) | 1.0 |
| IB-p-20 | 100000 | 15 | 250 | (6, ) | (3, ) | 1.0 |
| IB-p-25 | 100000 | 15 | 250 | (6, ) | (3, ) | 1.0 |
| IB-p-30 | 100000 | 15 | 250 | (6, ) | (3, ) | 1.0 |
| IB-p-35 | 100000 | 15 | 250 | (6, ) | (3, ) | 1.0 |
| IB-p-40 | 100000 | 15 | 250 | (6, ) | (3, ) | 1.0 |
| IB-p-45 | 100000 | 15 | 250 | (6, ) | (3, ) | 1.0 |
| IB-p-50 | 100000 | 15 | 250 | (6, ) | (3, ) | 1.0 |
| IB-p-55 | 100000 | 15 | 250 | (6, ) | (3, ) | 1.0 |
| IB-p-60 | 100000 | 15 | 250 | (6, ) | (3, ) | 1.0 |
| IB-p-65 | 100000 | 15 | 250 | (6, ) | (3, ) | 1.0 |
| IB-p-70 | 100000 | 15 | 250 | (6, ) | (3, ) | 1.0 |
| IB-p-75 | 100000 | 15 | 250 | (6, ) | (3, ) | 1.0 |
| IB-p-80 | 100000 | 15 | 250 | (6, ) | (3, ) | 1.0 |
| IB-p-85 | 100000 | 15 | 250 | (6, ) | (3, ) | 1.0 |
| IB-p-90 | 100000 | 15 | 250 | (6, ) | (3, ) | 1.0 |
| IB-p-95 | 100000 | 15 | 250 | (6, ) | (3, ) | 1.0 |
| IB-p-100 | 100000 | 15 | 250 | (6, ) | (3, ) | 1.0 |

*Table 6.* Industrial-Benchmark - Detailed information about collected dataset

# H. Task-Level Performance

| Task | Random Score | Expert Score | Vintix (Normalized) |
|---|---|---|---|
| assembly | 45,1 ± 3,3 | 297,9 ± 24,8 | 1.04 ± 0.10 |
| basketball | 3 ± 1,3 | 558,4 ± 69,3 | 1.02 ± 0.11 |
| bin-picking | 2 ± 0,6 | 424 ± 105,7 | 0.01 ± 0.01 |
| box-close | 78,7 ± 11,6 | 520 ± 122,6 | -0.04 ± 0.03 |
| button-press | 31,3 ± 4,3 | 642,9 ± 13 | 0.97 ± 0.07 |
| button-press-topdown | 31,6 ± 9,8 | 483,7 ± 41,7 | 0.94 ± 0.14 |
| button-press-topdown-wall | 30,8 ± 9 | 497,9 ± 31,2 | 0.97 ± 0.07 |
| button-press-wall | 10,3 ± 4,4 | 675,9 ± 15,2 | 1.00 ± 0.04 |
| coffee-button | 33,1 ± 7,8 | 731,4 ± 29,1 | 1.00 ± 0.06 |
| coffee-pull | 4,3 ± 0,5 | 294 ± 103,6 | 0.07 ± 0.20 |
| coffee-push | 4,3 ± 0,7 | 564,8 ± 106,7 | 1.01 ± 0.20 |
| dial-turn | 35,8 ± 42,8 | 788,4 ± 82,4 | 0.99 ± 0.07 |
| disassemble | 40,6 ± 6,8 | 527,3 ± 64,6 | 1.00 ± 0.12 |
| door-close | 5,5 ± 1,4 | 541,9 ± 25,6 | 1.02 ± 0.05 |
| door-lock | 115,7 ± 30,1 | 805,5 ± 42,7 | 0.35 ± 0.36 |
| door-open | 58,9 ± 10,1 | 587,6 ± 18,4 | 0.99 ± 0.05 |
| door-unlock | 99,8 ± 17,8 | 803,1 ± 19,6 | 0.21 ± 0.26 |
| drawer-close | 147,4 ± 291,1 | 868,4 ± 3,3 | 0.96 ± 0.01 |
| drawer-open | 126,3 ± 23,3 | 492,9 ± 2,5 | 1.00 ± 0.01 |
| faucet-close | 260,9 ± 26,4 | 755,4 ± 20,8 | 0.99 ± 0.03 |
| faucet-open | 242,7 ± 24,5 | 746,9 ± 12,9 | 0.97 ± 0.03 |
| hammer | 93,9 ± 9,7 | 687,6 ± 49,9 | 0.97 ± 0.12 |
| hand-insert | 2,7 ± 0,8 | 738,2 ± 36,6 | 0.19 ± 0.20 |
| handle-press | 73,6 ± 76,1 | 812,8 ± 170,8 | 1.00 ± 0.15 |
| handle-press-side | 59,4 ± 26,8 | 860,3 ± 32,8 | 0.99 ± 0.02 |
| handle-pull | 9,5 ± 6,5 | 703,3 ± 15,2 | 1.00 ± 0.02 |
| handle-pull-side | 2,1 ± 0,5 | 495,3 ± 49,3 | 1.00 ± 0.11 |
| lever-pull | 66,9 ± 16,9 | 575,5 ± 67,9 | 0.96 ± 0.16 |
| peg-insert-side | 1,8 ± 0,3 | 330,1 ± 159,4 | 1.01 ± 0.40 |
| peg-unplug-side | 4,6 ± 2,5 | 528,8 ± 87,1 | 0.75 ± 0.34 |
| pick-out-of-hole | 8,6 ± 49,9 | 401,1 ± 105,8 | 0.91 ± 0.20 |
| pick-place | 1,5 ± 0,9 | 423,6 ± 97,5 | 0.98 ± 0.22 |
| pick-place-wall | 0 ± 0 | 515,4 ± 162,5 | 1.04 ± 0.16 |
| plate-slide | 76,5 ± 12 | 531,1 ± 152 | 0.99 ± 0.34 |
| plate-slide-back | 33,8 ± 9,5 | 719,5 ± 88 | 0.99 ± 0.16 |
| plate-slide-back-side | 35,9 ± 13 | 727,5 ± 69,5 | 0.99 ± 0.09 |
| plate-slide-side | 22,6 ± 14 | 693,6 ± 82,4 | 0.83 ± 0.22 |
| push | 5,8 ± 2,6 | 747,7 ± 61,5 | 0.97 ± 0.14 |
| push-back | 1,2 ± 0,1 | 396,2 ± 109,3 | 0.95 ± 0.29 |
| push-wall | 6,5 ± 4 | 749,7 ± 11,7 | 0.99 ± 0.02 |
| reach | 154 ± 52,8 | 679,6 ± 131,8 | 1.01 ± 0.26 |
| reach-wall | 149 ± 35,7 | 748,2 ± 102,1 | 0.98 ± 0.17 |
| shelf-place | 0 ± 0 | 266,5 ± 30 | 1.01 ± 0.11 |
| soccer | 5,3 ± 3,9 | 509,3 ± 136,1 | 0.80 ± 0.35 |
| stick-pull | 2,8 ± 1,4 | 529,2 ± 20,9 | 0.92 ± 0.16 |
| stick-push | 2,9 ± 1,1 | 626 ± 46,9 | 0.90 ± 0.28 |
| sweep | 12,4 ± 8,5 | 518,4 ± 48 | 0.94 ± 0.12 |
| sweep-into | 16,9 ± 14,3 | 799 ± 14,9 | 1.00 ± 0.02 |
| window-close | 58,2 ± 15 | 591,1 ± 38,6 | 0.95 ± 0.17 |
| window-open | 42,9 ± 7,6 | 594,9 ± 56,2 | 0.96 ± 0.17 |

*Table 7.* Random, expert, and Vintix scores for Meta-World domain. Note that, here, we reported scores as in comparison to JAT model.

| Task | Random Score | Expert Score | Vintix (Normalized) |
|------|------|------|------|
| Ant-v4 | -459,2 ± 824,1 | 6368,2 ± 593,8 | 0,98 ± 0,10 |
| HalfCheetah-v4 | -299,8 ± 74,4 | 7782,8 ± 109,2 | 0,93 ± 0,03 |
| Hopper-v4 | 16,2 ± 8,4 | 3237,8 ± 707,8 | 0,86 ± 0,19 |
| Humanoid-v4 | 116,5 ± 31,6 | 7527,5 ± 38,8 | 0,97 ± 0,00 |
| HumanoidStandup-v4 | 37285,5 ± 3178 | 300990,1 ± 47970,1 | 1,02 ± 0,21 |
| InvertedDoublePendulum-v4 | 56,2 ± 16,1 | 9332,4 ± 498,1 | 0,65 ± 0,47 |
| InvertedPendulum-v4 | 5,6 ± 2,1 | 1000 ± 0 | 1,00 ± 0,00 |
| Pusher-v5 | -151,9 ± 8 | -40,1 ± 7 | 1,02 ± 0,08 |
| Reacher-v4 | -41,7 ± 3,4 | -5,6 ± 2,6 | 0,98 ± 0,07 |
| Swimmer-v4 | 3 ± 11,2 | 95,5 ± 3,6 | 0,98 ± 0,06 |
| Walker2d-v4 | 3,4 ± 6,4 | 5349,7 ± 254,6 | 1,00 ± 0,02 |

*Table 8.* Random, expert, and Vintix scores for the MuJoCo domain. Note that, here, we reported scores as in comparison to JAT model.

| Task | Random Score | Expert Score | Vintix (Normalized) |
|------|------|------|------|
| shadowhandblockstack | 95,6 ± 16 | 285 ± 41,8 | 1.17 ± 0.23 |
| shadowhandbottlecap | 110,6 ± 17,6 | 399,9 ± 57,6 | 0.81 ± 0.25 |
| shadowhandcatchabreast | 1,1 ± 0,6 | 65,6 ± 14,2 | 0.17 ± 0.32 |
| shadowhandcatchover2underarm | 4,9 ± 0,7 | 34,2 ± 6,1 | 0.92 ± 0.24 |
| shadowhandcatchunderarm | 1,7 ± 0,6 | 25,1 ± 6,1 | 0.72 ± 0.39 |
| shadowhanddoorcloseinward | 1,2 ± 0,5 | 8,8 ± 0,2 | 0.36 ± 0.02 |
| shadowhanddoorcloseoutward | 941,2 ± 43,8 | 1377,5 ± 15,7 | -1.27 ± 0.01 |
| shadowhanddooropeninward | -4,4 ± 41,3 | 409,9 ± 3,2 | 0.29 ± 0.02 |
| shadowhanddooropenoutward | 20,4 ± 40,9 | 617,1 ± 4,4 | 0.13 ± 0.02 |
| shadowhandgraspandplace | 6,8 ± 1,6 | 498,1 ± 51,1 | 0.97 ± 0.18 |
| shadowhandkettle | -191,9 ± 14,8 | 54,5 ± 4,2 | -0.99 ± 0.00 |
| shadowhandliftunderarm | -42,7 ± 8,4 | 404 ± 10,1 | 0.95 ± 0.03 |
| shadowhandover | 2,7 ± 0,7 | 34,6 ± 5,8 | 0.95 ± 0.03 |
| shadowhandpen | 4,5 ± 2,4 | 186,3 ± 19,5 | 0.52 ± 0.44 |
| shadowhandpushblock | 224,7 ± 66,5 | 457,4 ± 3,6 | 0.98 ± 0.01 |
| shadowhandreorientation | 127,1 ± 434,9 | 3040,1 ± 1986,8 | 0.89 ± 0.66 |
| shadowhandscissors | -23,3 ± 16,8 | 735,6 ± 24,8 | 1.03 ± 0.01 |
| shadowhandswingcup | -414,1 ± 27,9 | 3937,9 ± 601,1 | 0.95 ± 0.13 |
| shadowhandswitch | 50,4 ± 12,5 | 281,3 ± 1,1 | 0.95 ± 0.01 |
| shadowhandtwocatchunderarm | 2,2 ± 1,1 | 24,8 ± 6 | 0.03 ± 0.03 |

*Table 9.* Random and expert scores for Bi-DexHands domain

| Task | Random Score | Expert Score | Vintix (Normalized) |
|---|---|---|---|
| industrial-benchmark-0-v1 | $-348,9 \pm 32,7$ | $-180,7 \pm 3,1$ | $0.94 \pm 0.13$ |
| industrial-benchmark-5-v1 | $-379,7 \pm 59$ | $-193,8 \pm 2,3$ | $1.00 \pm 0.02$ |
| industrial-benchmark-10-v1 | $-395,4 \pm 68,9$ | $-215,1 \pm 2$ | $1.01 \pm 0.01$ |
| industrial-benchmark-15-v1 | $-424,4 \pm 83$ | $-229,8 \pm 4$ | $1.01 \pm 0.02$ |
| industrial-benchmark-20-v1 | $-455,8 \pm 88,2$ | $-249,4 \pm 2,2$ | $0.95 \pm 0.11$ |
| industrial-benchmark-25-v1 | $-453,6 \pm 78,8$ | $-272,4 \pm 5,5$ | $0.95 \pm 0.11$ |
| industrial-benchmark-30-v1 | $-480,3 \pm 76,6$ | $-288,3 \pm 5,4$ | $0.90 \pm 0.10$ |
| industrial-benchmark-35-v1 | $-508,5 \pm 87$ | $-314,1 \pm 6,7$ | $1.00 \pm 0.03$ |
| industrial-benchmark-40-v1 | $-515,8 \pm 77,4$ | $-337,8 \pm 8,8$ | $0.99 \pm 0.05$ |
| industrial-benchmark-45-v1 | $-543,3 \pm 85,1$ | $-360,9 \pm 7,4$ | $0.97 \pm 0.04$ |
| industrial-benchmark-50-v1 | $-574,9 \pm 69,8$ | $-377,9 \pm 7$ | $0.91 \pm 0.09$ |
| industrial-benchmark-55-v1 | $-597,6 \pm 69,9$ | $-402,1 \pm 6$ | $0.99 \pm 0.01$ |
| industrial-benchmark-60-v1 | $-625,2 \pm 83,4$ | $-430,3 \pm 4,8$ | $0.98 \pm 0.02$ |
| industrial-benchmark-65-v1 | $-649,6 \pm 62,2$ | $-449,8 \pm 4,1$ | $0.86 \pm 0.04$ |
| industrial-benchmark-70-v1 | $-691,7 \pm 87$ | $-471,1 \pm 4,2$ | $0.95 \pm 0.03$ |
| industrial-benchmark-75-v1 | $-717,2 \pm 72,3$ | $-474,3 \pm 3,2$ | $0.99 \pm 0.01$ |
| industrial-benchmark-80-v1 | $-757,8 \pm 105,9$ | $-485,4 \pm 3,1$ | $0.96 \pm 0.01$ |
| industrial-benchmark-85-v1 | $-812,8 \pm 154,9$ | $-507,6 \pm 2,9$ | $0.98 \pm 0.01$ |
| industrial-benchmark-90-v1 | $-846,4 \pm 132,7$ | $-522 \pm 3,6$ | $0.97 \pm 0.01$ |
| industrial-benchmark-95-v1 | $-895,5 \pm 146,5$ | $-545,8 \pm 3,3$ | $0.97 \pm 0.01$ |
| industrial-benchmark-100-v1 | $-986 \pm 199$ | $-561,8 \pm 4,6$ | $0.97 \pm 0.01$ |

*Table 10.* Random and expert scores for Industrial Benchmark domain

## I. Comparison with other cross-domain agents

| Task | Vintix | REGENT | JAT (Full Dataset) |
|---|---|---|---|
| Ant | $0.98 \pm 0.10$ | 0.17 | $0.88 \pm 0.29$ |
| HalfCheetah | $0.93 \pm 0.03$ | 0.34 | $0.89 \pm 0.03$ |
| Hopper | $0.86 \pm 0.19$ | 0.12 | $0.76 \pm 0.21$ |
| Humanoid | $0.97 \pm 0.00$ | 0.02 | $0.10 \pm 0.02$ |
| HumanoidStandup | $1.02 \pm 0.21$ | 0.26 | $0.35 \pm 0.09$ |
| InvertedDoublePendulum | $0.65 \pm 0.47$ | 0.02 | $0.93 \pm 0.14$ |
| InvertedPendulum | $1.00 \pm 0.00$ | 0.06 | $0.24 \pm 0.05$ |
| Pusher | $1.02 \pm 0.08$ | 0.90 | $1.00 \pm 0.05$ |
| Reacher | $0.98 \pm 0.07$ | 0.90 | $0.99 \pm 0.06$ |
| Swimmer | $0.98 \pm 0.06$ | 0.82 | $1.02 \pm 0.04$ |
| Walker2d | $1.00 \pm 0.02$ | 0.05 | $0.95 \pm 0.18$ |

*Table 11.* Performance of Vintix, REGENT and JAT on MuJoCo domain tasks.

| Task | Vintix | REGENT | JAT (Full Dataset) |
|---|---|---|---|
| assembly | $1.04 \pm 0.10$ | 0.83 | $0.96 \pm 0.17$ |
| basketball | $1.02 \pm 0.11$ | 0.68 | $-0.00 \pm 0.00$ |
| bin-picking | $0.01 \pm 0.01$ | 0.67 | $0.47 \pm 0.52$ |
| box-close | $-0.04 \pm 0.03$ | 0.93 | $0.89 \pm 0.39$ |
| button-press | $0.97 \pm 0.07$ | 0.62 | $0.86 \pm 0.30$ |
| button-press-topdown | $0.94 \pm 0.14$ | 0.62 | $0.51 \pm 0.17$ |
| button-press-topdown-wall | $0.97 \pm 0.07$ | 0.62 | $0.53 \pm 0.19$ |
| button-press-wall | $1.00 \pm 0.04$ | 0.94 | $0.94 \pm 0.18$ |
| coffee-button | $1.00 \pm 0.06$ | 0.84 | $0.38 \pm 0.41$ |
| coffee-pull | $0.07 \pm 0.20$ | 0.62 | $0.14 \pm 0.27$ |
| coffee-push | $1.01 \pm 0.20$ | 0.18 | $0.30 \pm 0.44$ |
| dial-turn | $0.99 \pm 0.07$ | 0.83 | $0.95 \pm 0.16$ |
| disassemble | $1.00 \pm 0.12$ | 2.24 | $0.17 \pm 3.91$ |
| door-close | $1.02 \pm 0.05$ | 1.00 | $0.99 \pm 0.06$ |
| door-lock | $0.35 \pm 0.36$ | 0.85 | $0.84 \pm 0.28$ |
| door-open | $0.99 \pm 0.05$ | 0.98 | $0.99 \pm 0.10$ |
| door-unlock | $0.21 \pm 0.26$ | 0.90 | $0.95 \pm 0.13$ |
| drawer-close | $0.96 \pm 0.01$ | 1.00 | $0.64 \pm 0.30$ |
| drawer-open | $1.00 \pm 0.01$ | 0.96 | $0.98 \pm 0.10$ |
| faucet-close | $0.99 \pm 0.03$ | 0.53 | $0.23 \pm 0.18$ |
| faucet-open | $0.97 \pm 0.03$ | 0.99 | $0.70 \pm 0.37$ |
| hammer | $0.97 \pm 0.12$ | 0.95 | $0.96 \pm 0.15$ |
| hand-insert | $0.19 \pm 0.20$ | 0.82 | $0.93 \pm 0.25$ |
| handle-press | $1.00 \pm 0.15$ | 0.99 | $0.84 \pm 0.32$ |
| handle-press-side | $0.99 \pm 0.02$ | 0.99 | $0.01 \pm 0.09$ |
| handle-pull | $1.00 \pm 0.02$ | 0.48 | $0.83 \pm 0.25$ |
| handle-pull-side | $1.00 \pm 0.11$ | 0.48 | $0.50 \pm 0.49$ |
| lever-pull | $0.96 \pm 0.16$ | 0.19 | $0.40 \pm 0.43$ |
| peg-insert-side | $1.01 \pm 0.40$ | 0.70 | $0.81 \pm 0.51$ |
| peg-unplug-side | $0.75 \pm 0.34$ | 0.31 | $0.17 \pm 0.32$ |
| pick-out-of-hole | $0.91 \pm 0.20$ | 0.01 | $0.00 \pm 0.00$ |
| pick-place | $0.98 \pm 0.22$ | 0.99 | $0.32 \pm 0.48$ |
| pick-place-wall | $1.04 \pm 0.16$ | 0.99 | $0.10 \pm 0.29$ |
| plate-slide | $0.99 \pm 0.34$ | 1.00 | $0.90 \pm 0.42$ |
| plate-slide-back | $0.99 \pm 0.16$ | 1.00 | $0.24 \pm 0.00$ |
| plate-slide-back-side | $0.99 \pm 0.09$ | 1.00 | $0.96 \pm 0.17$ |
| plate-slide-side | $0.83 \pm 0.22$ | 0.99 | $0.16 \pm 0.04$ |
| push | $0.97 \pm 0.14$ | 0.84 | $0.94 \pm 0.21$ |
| push-back | $0.95 \pm 0.29$ | 0.84 | $0.97 \pm 1.29$ |
| push-wall | $0.99 \pm 0.02$ | 0.81 | $0.21 \pm 0.30$ |
| reach | $1.01 \pm 0.26$ | 0.99 | $0.34 \pm 0.32$ |
| reach-wall | $0.98 \pm 0.17$ | 0.99 | $0.81 \pm 0.37$ |
| shelf-place | $1.01 \pm 0.11$ | 0.96 | $0.38 \pm 0.46$ |
| soccer | $0.80 \pm 0.35$ | 0.61 | $0.77 \pm 0.44$ |
| stick-pull | $0.92 \pm 0.16$ | 0.88 | $0.92 \pm 0.23$ |
| stick-push | $0.90 \pm 0.28$ | 0.75 | $0.48 \pm 0.48$ |
| sweep | $0.94 \pm 0.12$ | 0.91 | $0.01 \pm 0.04$ |
| sweep-into | $1.00 \pm 0.02$ | 0.91 | $0.99 \pm 0.06$ |
| window-close | $0.95 \pm 0.17$ | 0.90 | $0.67 \pm 0.39$ |
| window-open | $0.96 \pm 0.17$ | 0.97 | $0.98 \pm 0.12$ |

*Table 12.* Performance of Vintix, REGENT and JAT on Meta-World domain tasks.

