# OpenReview forum: "Vintix: Action Model via In-Context Reinforcement Learning"
_ICML.cc/2025/Conference — ICML 2025 poster_

### Official Review · Reviewer_LDFi · 2025-03-03

**Overall Recommendation:** 3

**Summary:**

The paper introduces Vintix, a cross-domain action model capable of in-context reinforcement learning (ICRL).

The key contributions include:

(1) Continuous Noise Distillation, extending existing work to continuous action spaces;

(2) a cross-domain dataset spanning 87 tasks across 4 environments (Meta-World, MuJoCo, Bi-DexHands, Industrial-Benchmark); and

(3) empirical evidence showing the model's ability to self-correct during inference through many-shot ICRL. The authors demonstrate that their model outperforms previous approaches like JAT on Meta-World (+32%) and MuJoCo (+13.5%), while also showing some ability to adapt to parametric variations in environments.

**Claims And Evidence:**

The core claim about ICRL, self-correction capabilities are supported by convincing experiments, and showing good performance with increasing context.

And the model structure

**Essential References Not Discussed:**

This paper lacks discussion of sequence modeling approaches to RL like decision transformer, trajectory transformer and other works.

**Ethics Expertise Needed:**

["Other expertise"]

**Experimental Designs Or Analyses:**

The experimental design generally sound. The authoers use appropriate metrics and provide confidence intervals. The staged approach to evaluation (self-correction on training tasks, comparison to baselines, generalization to parametric variations and new tasks) makes sense.

However, this work lack ablation studies or discussions about dataset quality, cumulative learning and task diversity. Authors should discuss the these three key notes. I think it could be pretty important for future works and can certificant current findings in RL and LLM.

**Methods And Evaluation Criteria:**

The methods are appropriate for investigating in-context adaptation. The Continuous Noise Distillation technique is a sensible extension to continuous domains, and the benchmark environments are standard in the field. However, the evaluation could be strengthened by comparing against more contemporary baselines beyond JAT, and by more rigorously analyzing the quality of the collected datasets and their impact on performance.

**Other Comments Or Suggestions:**

The paper would benefit from a clearer discussion of the limitations of the approach, particularly regarding generalization. A more rigorous analysis of what constitutes "learning" in this context would strengthen the theoretical foundation. I think this work more like gato.

**Other Strengths And Weaknesses:**

Strengths:

The cross-domain approach is ambitious and moves beyond single-domain studies common in previous work.
The open-sourced datasets could be valuable to the community.

Weaknesses:

The paper lacks sufficient analysis of dataset quality and its impact on performance.
Limited generalization to truly new tasks raises questions about whether the model is actually learning adaptation strategies or merely memorizing task-specific behaviors.

**Questions For Authors:**

How would you distinguish the learning happening in your approach from standard supervised learning on expert trajectories? What evidence suggests that your model is actually performing reinforcement learning rather than just mimicking behavior?
Can you provide more detailed analysis of the quality of the datasets used for training and how this affects performance? Was there any filtering of suboptimal demonstrations?
The limited generalization to new tasks seems problematic for a method focused on adaptation. What modifications might improve zero-shot transfer to entirely new tasks?

**Relation To Broader Scientific Literature:**

This work builds upon Algorithm Distillation (Laskin et al., 2022) and extends it to multiple domains and continuous action spaces. It relates to the broader trend of in-context learning in LLMs, applying similar principles to RL. The paper properly situates itself relative to related work in Meta-RL (both memory-based approaches like Duan et al. and offline methods), generalist agents like Gato, and learned optimizers.

**Theoretical Claims:**

No formal theoretical claims or proofs are presented in the paper.

---

> ### Author Rebuttal · Authors · 2025-03-31
>
> Thank you for your review. Based on your feedback, we have identified these key points:
>
> 1. **How does Vintix compare to the latest action models?**
> 2. **Does Vintix learn to adapt or does it mimic expert through task identification?**
> 3. **How does the quality of dataset impact performance?**
> 4. **Providing more detailed supplementary materials.**
>
> We include responses and experiments conducted within the rebuttal period. If anything is missing, feel free to reach out — we will respond as promptly as possible
>
> ---
>
> ### **How does Vintix compare to the latest action models?**
>
> To broaden the comparison beyond JAT, we include expert-normalized scores for REGENT on MuJoCo and MetaWorld — the two domains shared across all works ([Table 1](https://postimg.cc/1n863r8d), [Table 2](https://postimg.cc/FfhhDCrt))
>
> Vintix outperforms both JAT and REGENT in normalized scores but lags behind REGENT on entirely unseen tasks. Notably, unlike REGENT, Vintix received no demonstrations for these tasks and operated in a fully cold-start setting.
>
> ---
>
> ### **Does Vintix learn to adapt or does it mimic expert through task identification?**
>
> **Exp 1**
>
> We evaluated whether AD-style training on noise-distilled trajectories outperforms behavioral cloning with context, using a model identical to Vintix but trained solely on expert data. Both models were evaluated using the procedure described in lines 238–239 of the paper.
>
> [Figure 1](https://postimg.cc/XG777HmK) shows that ED converges to ~0.8, while AD reaches 0.97 (MuJoCo) and 0.95 (MetaWorld). This highlights the importance of policy-improvement data for self-correcting behavior. ED shows stable performance in MuJoCo and surprisingly improves with more shots in MetaWorld, likely due to task inference still occuring under a shared encoder (requiring 4 episodes in context) but still underperforms AD despite being trained on the same amount of data.
>
> **Exp 2**
>
> Secondly, we examine whether Vintix relies on the reward signal for self-improvement during inference. To test this, we re-trained Vintix without access to rewards and compared it to the original model using the cold-start inference on tasks from the MuJoCo and MetaWorld domains.
>
> [Figure 1](https://postimg.cc/XG777HmK) shows that reward feedback is crucial: the masked-reward variant performs worse asymptotically in both domains and converges more slowly on MetaWorld. These results suggest that training on data reflecting policy improvement is essential for enabling in-context reinforcement learning
>
> ---
>
> ### **How does the quality of dataset impact performance?**
>
> To evaluate the impact of dataset quality, we collected a MuJoCo dataset using an untuned noise decay schedule, which led to non-smooth policy improvement on some tasks. We re-trained the AD model on both datasets and compared them using the cold-start evaluation
>
> [Figure 2](https://postimg.cc/HVQGR0PT) shows that certain tasks were strongly affected by poor decay scheduling, while others remained stable. [Figure 3](https://postimg.cc/xXtZV6GV) indicates that models trained on lower-quality data exhibit weaker asymptotic performance
>
> Despite reaching an expert-normalized score of 0.81, these findings highlight the importance of using high-quality data with smooth, progressive improvement to maximize performance
>
> ---
>
> ### **Providing more detailed supplementary materials**
>
> We provide a link to an [anonymous repo](https://anonymous.4open.science/r/vintix-rebuttal-icml-2025-7F33 ) with Vintix code and [training dataset]( https://tinyurl.com/426ckafn). Extra supplementary material is available in paper’s appendix
>
> ---
>
> **Discussion of sequence based RL approaches** With the rise of Transformers for modeling sequential data, several works ([1](https://arxiv.org/abs/2106.01345), [2](https://arxiv.org/abs/2106.02039)) formulated MDP as a causal sequence modeling problem. ([1](https://arxiv.org/abs/2106.01345)) focused on reward conditioning treating each MDP element as a separate token, while ([2](https://arxiv.org/abs/2106.02039)) applied beam search over discretized SAR tuples.
>
> Subsequent research has expanded this area by making models that maximize returns ([3](https://arxiv.org/abs/2405.08740)), adapting DT to online learning ([4](https://arxiv.org/abs/2202.05607)), and replacing the Transformer with SSM backbones like Mamba ([5](https://arxiv.org/abs/2406.00079)).
>
> **On suboptimal demonstrators** We did not filter out failed expert trajectories to avoid biasing the dataset, particularly in cases where failures may be correlated. Instead, we addressed noisy behavior by further training the demonstrators.
>
> **On generalization to unseen tasks**
>
> 1. *Scaling the dataset* AD benefits from a large number of tasks to generalize effectively. We plan to expand the dataset with new domains.
> 2. *Domain-invariant architecture* Using VLA-like models to map modalities into a shared embedding space may improve cross-domain transfer and reduce reliance on task identification.

---

### Official Review · Reviewer_bYMy · 2025-03-14

**Overall Recommendation:** 3

**Summary:**

This paper explores the potential of In-Context Reinforcement Learning (ICRL) for developing generalist agents capable of learning and adapting through trial-and-error interactions at inference time. The authors present Vintix, a fixed, cross-domain action model that leverages the Algorithm Distillation (AD) framework to learn behaviors across various tasks. Vintix uses Continuous Noise Distillation to collect training data from multiple domains. The model demonstrates significant self-correction capabilities, achieving near demonstrator-level performance on multiple training tasks and adapting to parametric variations at inference time.

**Claims And Evidence:**

Clear.

**Essential References Not Discussed:**

None.

**Experimental Designs Or Analyses:**

The chosen tasks for experimental validation are overly standardized and fail to encompass a broad spectrum of extreme or unconventional scenarios. This narrow focus could severely limit the applicability of the method in complex real-world environments.

**Methods And Evaluation Criteria:**

Yes.

**Other Comments Or Suggestions:**

None.

**Other Strengths And Weaknesses:**

Strengths:
	1.	This paper proposes a novel data collection strategy, which incrementally reduces the uniform noise injected into demonstrator policies. This extended mechanism enhances data collection efficiency and makes algorithm distillation more feasible in reward-oriented reinforcement learning.
	2.	Using an improved transformer architecture for model training, detailed optimizations ensure efficient learning and adaptability. This standardized training process enhances the model's generalization and applicability.
	3.	Experimental results demonstrate that the Vintix model can self-correct using contextual information at inference time, progressively reaching near-demonstrator levels. It shows strong cross-domain generalization capability.
	4.	The paper constructs a large cross-domain dataset covering 87 tasks across 4 domains.

Weaknesses：
	1.	The chosen tasks for experimental validation are overly standardized and fail to encompass a broad spectrum of extreme or unconventional scenarios. This narrow focus could severely limit the applicability of the method in complex real-world environments.
	2.	The paper fails to clearly elucidate the practical advantages of Continuous Noise Distillation in real-world scenarios. Besides, the paper lacks detailed discussion regarding the potential impact and value of this method in tangible applications.
	3.	The dataset construction and utilization process is complex and potentially cumbersome, posing substantial challenges for practical implementation. This intricate dependency on specific steps and environments restricts the generalizability and reproducibility of the method.

**Questions For Authors:**

None.

**Relation To Broader Scientific Literature:**

None.

**Theoretical Claims:**

Yes.

---

> ### Author Rebuttal · Authors · 2025-03-31
>
> Thank you for the review. Based on your feedback, we believe the central point of discussion is the practicality — in a broad, real-world sense — of the proposed approach. In particular, we address the following raised topics:
>
> - **Do the selected domains represent a broad spectrum of challenging, real-world-relevant tasks?**
> - **Does the proposed data collection method offer advantages over existing approaches (e.g., learning histories from PPO or optimal action labels as in DPT)?**
>
> While our work focuses on simulated environments and does not yet make claims about real-world transfer, we argue below that both our choice of domains and our data collection methods are well-justified. Moreover, we believe they provide a strong foundation for the continued development of action models within the framework of in-context reinforcement learning.
>
> ---
>
> ### **Do the selected domains encompass a broad spectrum of difficult and real-world tasks?**
>
> > The chosen tasks for experimental validation are overly standardized and fail to encompass a broad spectrum of extreme or unconventional scenarios. This narrow focus could severely limit the applicability of the method in complex real-world environments.
>
> When benchmarking models of this kind, it is important to balance standardized environments—which ensure reproducibility and fair comparison—with task suites that reflect real-world complexity. A central goal of our work is to contribute datasets and results to the broader community, laying a foundation for further scaling and development.
>
> This objective imposes constraints on domain selection: environments must be open-source and widely adopted by the research community. While MuJoCo is included as a field standard, the remaining domains were chosen for their practical relevance and the complexity of the challenges they present:
>
> - **Meta-World ML45**: A highly challenging benchmark where state-of-the-art online Meta-RL algorithms achieve a success rate of just 0.4 [(Shala et al., 2025)](https://openreview.net/forum?id=UENQuayzr1). It is widely used and practically motivated, with over 490 citations since 2024.
>
> - **Industrial Benchmark**: A synthetic suite that models industrial optimization problems with complex dynamics, heteroscedastic noise, delayed multi-objective rewards, and partial observability. **Notably, the benchmark was explicitly designed to test RL algorithms under conditions resembling real-world industrial control problems.**
>
> - **Bi-DexHands**: Grasping is a core challenge in robotics, critical for tasks in human-centric environments [(Billard et al., 2019)](https://www.science.org/doi/10.1126/science.aat8414). Despite extensive research, it remains difficult due to the high-dimensional action space. Bi-DexHands offers a diverse suite of grasping tasks, and is recognized as it was accepted to last year’s NeurIPS Benchmarks and Datasets track last year.
>
> ---
>
> ### **Does the proposed collection method offer advantages over the existing ones (e.g, learning histories from PPO or optimal action labels as in DPT)?**
>
> > The paper fails to clearly elucidate the practical advantages of Continuous Noise Distillation in real-world scenarios. Besides, the paper lacks detailed discussion regarding the potential impact and value of this method in tangible applications.
>
> Continuous Noise Distillation significantly simplifies dataset collection for Algorithm Distillation. In vanilla AD, a new RL agent must be trained for each learning history, often requiring millions or billions of steps. This process is time-consuming, unpredictable, and limits scalability.
>
> In contrast, Continuous Noise Distillation enables efficient, controllable data collection using only a demonstrator policy. Users can set the length of each learning history, reducing computational overhead and enhancing practicality in time- and cost-sensitive scenarios.
>
> > The dataset construction and utilization process is complex and potentially cumbersome, posing substantial challenges for practical implementation. This intricate dependency on specific steps and environments restricts the generalizability and reproducibility of the method.
>
> While dataset construction does require some effort, it remains relatively simple compared to other In-Context RL methods. It only needs demonstrator policies and environment access. By contrast, [AD](https://arxiv.org/abs/2210.14215) requires full RL learning histories, and [DPT](https://arxiv.org/abs/2306.14892) needs expert-provided target actions. Generalist agents like [JAT](https://arxiv.org/abs/2402.09844), [GATO](https://arxiv.org/abs/2205.06175), and [Baku](https://arxiv.org/abs/2406.07539) also rely on expert demonstrations. Although simplifying data collection for In-Context RL is an important direction, it is beyond the scope of this work.

---

### Official Review · Reviewer_5G9i · 2025-03-14

**Overall Recommendation:** 2

**Summary:**

This work explores In-Context Reinforcement Learning (ICRL) as a method for developing generalist agents that can learn through trial-and-error during inference. The proposed approach is built on Algorithm Distillation (AD), a prior in-context RL work. Specifically, based on AD, the authors adopt continuous noise distillation approach to construct the training datasets. In addition, instead of training on single domains, the proposed approach is trained on four different domains with a total of 87 tasks and 1.6M episodes. The results suggest that the proposed approach could learn an agent that can self-correct and improve its performance during test time across the four tested domains. The paper concludes that ICRL is a potential approach for creating scalable generalist decision-making systems.

## update after rebuttal
The authors' response has only partially addressed my concerns. I still believe that the paper's main contribution may not be substantial. Continuous noise distillation and cross-domain datasets may not be significant enough advancements. Additionally, the lack of comparison with the vanilla AD remains a concern. While I acknowledge the difficulty of collecting all training data in a short time frame, comparing with vanilla AD using a subset of tasks could still be insightful. Therefore, I keep my score unchanged (weak reject).

**Claims And Evidence:**

The reviewer found that some of the claims made in the paper lack sufficient evidentiary support. Notably, the paper does not provide experimental results demonstrating that the proposed approach outperforms the vanilla Algorithm Distillation (AD) method. This comparison is crucial to evaluating the effectiveness of the proposed approach.

Furthermore, certain claims lack concrete experimental validation. For instance, in Section 2.3.2, the authors assert that standardizing reward functions using task-specific factors significantly enhances model performance. However, this assertion is not substantiated by any empirical evidence in the presented work. The absence of such evidence makes it challenging to assess the validity of this claim.

To strengthen the paper, the authors should consider conducting additional experiments that directly compare the proposed approach with the vanilla AD method. This would provide concrete evidence regarding the relative performance of the two approaches. Additionally, the authors should provide empirical evidence to support claims regarding the impact of specific modifications, such as the standardization of reward functions, on model performance.

**Essential References Not Discussed:**

None

**Experimental Designs Or Analyses:**

The reviewer finds that the evaluation of the proposed approach is lacking, as there are no baseline comparisons included in the main experiments (Figure 4). The inclusion of baselines such as algorithm distillation [a], AMMAGO [b], and Decision-Pretrained Transformer [c] would enable a more thorough evaluation of the effectiveness of the proposed approach by providing a point of reference and comparison.

[a] In-context Reinforcement Learning with Algorithm Distillation, Laskin, 2022

[b] AMAGO: Scalable In-Context Reinforcement Learning for Adaptive Agents, Grigsby, 2024

[c] Supervised Pretraining Can Learn In-Context Reinforcement Learning, Lee, 2023

**Methods And Evaluation Criteria:**

The reviewer has several concerns regarding the technical contributions of the paper. The two primary technical components presented are continuous noise distillation and cross-domain dataset and training. The reviewer finds neither contribution to be sufficiently substantial. Continuous noise distillation is viewed as a minor extension of the discrete noise distillation introduced in Zisman et al. (2024a), differing only in the application of uniform random noise. The use of a cross-domain dataset and training, while potentially interesting, lacks sufficient novelty or complexity to be considered a significant contribution on its own.

**Other Comments Or Suggestions:**

Line 174 - 175 (left) reads “...has access only to the dimensionality-based group identifier, but not to an individual task identifier.” Please clarify what "group identifier" and "task identifier" mean, and provide examples of each to illustrate the distinction.

**Other Strengths And Weaknesses:**

Please see the above discussion.

**Questions For Authors:**

In the comparison with JAT, the authors mentioned that data collection for the proposed approach uses improved expert performance, which is not used in the baselines. Please explain why the same expert cannot be used for both the baseline and the proposed approach to ensure a fair comparison.

**Relation To Broader Scientific Literature:**

This paper is strongly related to algorithm distillation (AD) [a]. It applied AD to a cross-domain setting.

**Theoretical Claims:**

No formal theoretical claim is presented.

---

> ### Author Rebuttal · Authors · 2025-03-31
>
> Thank you for the time and effort you devoted to reviewing our paper. We have identified the following key issues for further discussion:
>
> **Is it possible to include experiments comparing the proposed approach with vanilla AD to provide stronger evidence?**
>
> Vintix diverges from standard AD through noise-distilled data methodology, replacing RL learning histories because:
> - [Zisman et al. (2024a)](https://arxiv.org/abs/2312.12275) link noise distillation to in-context learning emergence
> - It improves trajectory control by avoiding lengthy, sample-inefficient PPO histories, which demand large computational resources and exhibit unstable convergence
> - It permits low-cost dataset generation via open-source expert policies
>
> Vintix functions as a standard AD framework applied to cross-domain, noise-distilled datasets. Recollecting dataset for vanilla AD requires retraining 1157 (87 tasks * 13.3 trajectories) RL agents—computationally unfeasible for rebuttal timelines
>
> **What evidence might be presented to demonstrate that standardizing reward functions with task-specific factors positively influences model performance?**
>
> We conducted a hyperparameter-tuning experiment on the Humanoid task. Over 50 hyperparameter configurations were evaluated. [Figure 1](https://postimg.cc/PPVCyf8t) presents the results, showing that using a non-standard reward scale yielded better scores. Before training Vintix,we performed analogous sweeps on each task. We don’t claim that modifying reward scales always leads to better models,but observed performance gains under our chosen settings
>
> **What is the key novelty of the paper that makes its technical contribution sufficiently substantial?**
>
> This work advances beyond noise integration in distillation by offering empirical principles for selecting ε-decay functions. We show noise-augmented distillation enables dynamic self-correction during inference, particularly in tasks with complex dynamic, high-dimensional actions, and partial observability. Also, we have open-sourced collected datasets for In-Context RL. To our knowledge, it was previously undertaken by the JAT, and their dataset comprises expert trajectories only.
>
> Our model training prioritized challenging domains over simplistic ones. The MetaWorld ML45 benchmark, a prominent robotic manipulation suite with 490+ citations since 2024, presents significant difficulty: state-of-the-art online MetaRL methods achieve only 0.4 success rates ([Shala et al., 2025](https://openreview.net/pdf?id=UENQuayzr1)). Industrial Benchmark introduces synthetic tasks simulating industrial optimization challenges, characterized by complex dynamics, heteroscedastic noise, delayed rewards and partial observability. Bi-DexHands addresses robotic grasping (a critical capability for human-environment interactions) through tasks testing RL and MetaRL algorithms against high-dimensional action space limitations. To our knowledge, no prior work has applied methods beyond behavioral cloning on such challenging cross-domain datasets
>
> **What is the performance of ADε in comparison with other methods?**
>
> Comparison with other offline MetaRL approaches is indeed a valuable experiment. Vanilla AD is discussed in the first paragraph. AMAGO is an online off-policy MetaRL approach limited to a single domain, while Vintix is offline and trains on fixed data. AMAGO is both difficult to replicate and challenging to adapt to the offline setting. DPT is unable to learn in-context in partially observable MDPs (Appendix H of [Nikulin et al.(2024)](https://arxiv.org/abs/2406.08973)). Bi-DexHands and Industrial Benchmark are partially observable
>
> We implemented DPT and trained it on MuJoCo. We also trained Vintix only on MuJoCo, using the same transformerparameters. [Figure 2](https://postimg.cc/3WfFzTsw) shows that DPT’s training loss exceeded Vintix’s and displayed spikes. Validation followed the same procedure with an empty initial context. As shown in [Figure 3](https://postimg.cc/m14Ws9zQ),DPT performs poorly, necessitating further tuning.No prior work, to our knowledge,has applied DPT to MuJoCo tasks, leaving it for future study
>
> **What is the difference between "group identifier" and "task identifier"?**
>
> The task identifier is a unique ID assigned to each task in the dataset. The group identifier is an ID indicating whether a group of tasks shares the same observation and action spaces (in terms of dimensionality and the semantic meaning of each channel)
>
> **Why were some JAT experts retrained to collect the dataset?**
>
> During data collection, we assessed JAT demonstrations and found several low-performing experts (zero success rates). Prioritizing dataset quality and model evaluation against expert benchmarks, we retrained selected experts. However, fair comparison remained possible through score normalization against updated benchmarks, intentionally lowering Vintix’s scores relative to JAT. This imposed stricter evaluation conditions on our model

---

### Official Review · Reviewer_U2Cv · 2025-03-21

**Overall Recommendation:** 4

**Summary:**

This paper proposes a method to train a general ICRL-capable agent following a version of Algorithm distillation (aka noise distillation) across four environment suites (aka domains). Their model architecture (like JAT) makes a complete transition into one token allowing them to expand to larger contexts for ICRL. The trained agent, after it is run from a cold start (with nothing in the context) in the training environments and after it converges (in returns) after many episodes (or shots), results in a conditioned agent that can improve performance on training metaworld and mujoco tasks over JAT. Vintix also demonstrates generalization to parameter varaiations in the environments.

**Claims And Evidence:**

Pros:

* The work is very well written. It takes a simple idea (AD^{\epsilon}) and scales it to multiple domains for the first time in ICRL.
* The results on improved performance in both training environments and parametric variations of those is very interesting. In particular, with the advent of test-time compute, this work on ICRL appears even more timely.
* The admission of only early signs of ICRL in new unseen tasks is a great pro!
* The paper's claims are supported by adequate evidence.

Cons:

* I am not certain if it is the adaptability of Vintix that improves performance over JAT in the trained environments or the context it is conditioned on before it's score is calculated. Like the inference time LLM works, is there a way to control the amount of time Vintix takes to converge to identify if taking longer results in better performance (because of a larger conditioning context)?
* It does appear that all domains do not have image-based observations. Do the authors have a way to scale generalization to new environments with parameteric variations when the observations have images? Does this require expert demonstrations like that seen in ICIL methods (like REGENT)?

**Essential References Not Discussed:**

NA

**Experimental Designs Or Analyses:**

Yes, I checked all experimental analyses. They are sounds. Please see "Claims And Evidence" for the detailed pros and cons.

**Methods And Evaluation Criteria:**

Yes. But, I have raised a couple of questions on evaluation in the cons above.

**Other Comments Or Suggestions:**

NA

**Other Strengths And Weaknesses:**

Addressed in "Claims And Evidence".

**Questions For Authors:**

Please see cons in "Claims And Evidence"

**Relation To Broader Scientific Literature:**

This work improves the frontier of ICRL. While some works in ICIL like REGENT have shown generalization to new environments, they do require few expert trajectories to retrieve a suitable context from. This work, building on ICRL methods like AD and AD^{\epsilon} (and going against the trend of ICRL methods like DPT) demonstrates early signs of success in generalization to new tasks and strong signs of generalization to parametric variations of tasks/envs. The ability to modulate the amount of ICRL until convergence would allow for a sort of test-time scaling here that would be great (if the authors can do something like that).

**Theoretical Claims:**

No proofs.

---

> ### Author Rebuttal · Authors · 2025-03-31
>
> Thank you for the review. Based on your feedback, we believe the main points of discussion can be distilled into the following:
>
> - **Does Adaptability (or inference-time learning) happen, or is it just Task Identification (context conditioning)?**
> - **How to scale to non-vector-based/proprioceptive modalities, e.g., images?**
>
> Below, we present our responses and the experiments we conducted during the limited rebuttal phase. If there’s anything we’ve overlooked or if further clarification is needed, please let us know—we’ll respond promptly within the available timeframe.
>
> ---
>
> ### **Does Adaptability (or Inference-Time Learning) Happen, or Is It Just Task Identification (Context Conditioning)?**
>
> You’re right to highlight the challenge of disentangling adaptability (i.e., inference-time learning) from task identification via conditioning. This is a recurring issue in meta-learning research, especially when attempting to distinguish in-weights learning from in-context learning at scale.
>
> To ensure that our results are not simply a consequence of task identification — and to explore the trade-off between in-context learning and in-weights learning — we perform several key ablations of our proposed approach:
>
> - **Algorithm Distillation (AD) with Masked Rewards** — The model is trained on noise-induced improvement trajectories, but with rewards masked out during training.
> - **Expert Distillation (ED)** — The model is trained exclusively on expert-level trajectories, similar to JAT/GATO-style behavior cloning.
>
> These ablations help shed light on the contribution of improvement trajectories and reward signals, thereby going beyond task identification and supporting the case for learning dynamics.
>
> ---
>
> **Vintix vs. Expert Distillation**
>
> We trained a model with the same architecture as Vintix (transformer backbone, encoders, loss function), but only on expert demonstrations (ED). We evaluated both ED and Vintix using the cold-start procedure (lines 238–239 in the paper).
>
> Results ([link](https://postimg.cc/N59JBH9n)) show that *Expert Distillation underperforms relative to AD* on both domains. ED reaches an average expert-normalized score of 0.8 on MuJoCo and Meta-World, while AD achieves 0.97 and 0.95, respectively.
>
> This suggests that the *structure of policy improvement* in the dataset is valuable for enabling self-correcting behavior and high performance. Notably, ED’s performance in MuJoCo stays flat across different shot counts, but improves in Meta-World as more episodes are provided.
>
> This implies ED partially learns task identification—especially challenging in Meta-World, where a shared encoder spans tasks. Our findings suggest ED requires ~4 episodes to infer the current task. However, *even after identifying the task, ED fails to reach AD’s asymptotic performance, despite having the same data volume*.
>
> ---
>
> **Vintix vs. Algorithm Distillation with No Rewards**
>
> In the second experiment, we aim to assess whether Vintix is utilizing both structure of policy improvement and the reward function—in other words, whether the reward signal contributes to the model's ability to self-improve during inference.
>
> To investigate this, we re-trained the Vintix model without access to rewards and compared its performance to the original version of Vintix (trained with rewards), using the previously described cold-start inference procedure on training tasks from the MuJoCo and Meta-World domains.
>
> The evaluation results ([link](https://postimg.cc/N59JBH9n)) suggest that the reward signal plays a role in achieving performance comparable to the demonstrator. *AD trained with masked rewards performs worse asymptotically across both domains and shows slower convergence on the Meta-World domain.*
>
> These findings indicate that reward feedback is essential for effective self-improvement during inference, supporting the view that supervised training on a dataset containing policy improvement and rewards enhances the model’s in-context reinforcement learning capabilities.
>
> ---
>
> ### **How to Scale to Non-Vector-Based/Proprioceptive Modalities (e.g., Images)?**
>
> Vintix currently operates on proprioceptive inputs, extending it to image-based observations is great direction for future work. To extend it to image-based observations with parametric variations, a natural approach would be to use a vision foundation model (e.g., BLIP) to encode visual input on the environment side.
>
> Variations such as object color or distractors can be applied prior to encoding and will be captured in the resulting image embeddings. These can then be passed to Vintix’s MLP encoder as n-dimensional inputs.
>
> This preserves the existing pipeline structure: only the image encoding is delegated to the environment. As before, training requires noise-distilled policy improvement data; inference remains unchanged.

---

> > ### Comment · Reviewer_U2Cv · 2025-04-06
> >
> > Thank you for the response. I will keep my score at 4: accept.

---

### Decision · Program_Chairs · 2025-05-01

**Decision:**

Accept (poster)

**Comment:**

This paper introduces an extension of a prior work on noise distillation for In-Context Reinforcement Learning (ICRL). The majority of the reviewers appreciated comprehensive evaluation across multiple continuous control domains, achieving strong performance on trained tasks and the potentially valuable contribution of its large-scale dataset generated via continuous noise distillation. There was a suggestion about the lack of comparison to vanilla AD. Although I understand AD is computationally expensive, I would encourage the authors to include comparison to AD at least on one task to show the relative strength. Assuming that this is reflected in the camera-ready version, I recommend to accept this paper.